# Scaling Up Multi-Task
# Robotic Reinforcement Learning

**Dmitry Kalashnikov**[*†]    **Jake Varley**[*†]    **Yevgen Chebotar**[†]    **Benjamin Swanson**[†]

**Rico Jonschkowski**[†]    **Chelsea Finn**[†]    **Sergey Levine**[†]    **Karol Hausman**[*†]

**Abstract:** General-purpose robotic systems must master a large repertoire of diverse skills. While reinforcement learning provides a powerful framework for acquiring individual behaviors, the time needed to acquire each skill makes the prospect of a generalist robot trained with RL daunting. We study how a large-scale collective robotic learning system can acquire a repertoire of behaviors simultaneously, sharing exploration, experience, and representations across tasks. In this framework, new tasks can be continuously instantiated from previously learned tasks improving overall performance and capabilities of the system. To this end, we develop a scalable and intuitive framework for specifying new tasks through user-provided examples of desired outcomes, devise a multi-robot collective learning system for data collection that simultaneously collects experience for multiple tasks, and develop a scalable and generalizable multi-task deep reinforcement learning method, which we call MT-Opt. We demonstrate how MT-Opt can learn a wide range of skills, including semantic picking (i.e., picking an object from a particular category), placing into various fixtures (e.g., placing a food item onto a plate). We train and evaluate our system on a set of 12 real-world tasks with data collected from 7 robots, and demonstrate the performance of our system both in terms of its ability to generalize to structurally similar new tasks, and acquire distinct new tasks more quickly by leveraging past experience. We recommend viewing the videos at https://karolhausman.github.io/mt-opt/

**Keywords:** Multi-Task Reinforcement Learning, Robot Learning

## 1   Introduction

Today's deep reinforcement learning (RL) methods, when applied to real-world robotic tasks, provide an effective but expensive way of learning skills [1, 2]. While existing methods are effective and able to generalize, they require considerable on-robot training time. For example, the QT-Opt [1] system can learn vision-based robotic grasping, but it requires over $500,000$ trials collected across multiple robots. While such sample complexity may be reasonable if the robot needs to perform a single task, it becomes costly if we consider the prospect of training a general-purpose robot with a large repertoire of behaviors, where each behavior is learned in isolation, starting from scratch. Can we instead *amortize* the cost of learning this repertoire over multiple skills, where the effort needed to learn whole repertoire is reduced compared to learning each skill in isolation?

Prior work suggests that multi-task RL can amortize the cost of single-task learning [3, 4, 5, 6, 7]. Insofar as the tasks share common structure, if that structure is discovered by the learning algorithm, all of the tasks can in principle be learned more efficiently than learning each task individually. In addition, by collecting experience simultaneously using controllers for a variety of tasks with different difficulty, the easier tasks can "bootstrap" the harder tasks. Finally, by enabling the multi-task RL policy to learn shared representations, learning new tasks can become easier over time as the system acquires more skills and learns more widely-useful aspects of the environment.

However, to realize these benefits for a real-world robotic learning system, we need to overcome a number of major challenges [8, 9, 10, 11], which have so far made it difficult to produce a large-scale demonstration of multi-task image-based RL in the real world. First, multi-task reinforcement

---

[*] Equal contribution

[†] Google Research, Robotics at Google Team

5th Conference on Robot Learning (CoRL 2021), London, UK.

learning is known to be exceedingly difficult from the optimization standpoint [8, 12]. Second, a real-world multi-task learning framework requires the ability to easily and intuitively define rewards for a large number of tasks. Third, while all task-specific data could be shared between all the tasks, it has been shown that reusing data from non-correlated tasks can be harmful to the learning process [13]. Lastly, to receive the benefits from shared, multi-task representation, we need to significantly scale up our algorithms, the number of tasks, and the robotic systems themselves.

The main contribution of this paper is a general multi-task learning system, which we call MT-Opt. MT-Opt realizes the hypothesized benefits of multi-task RL in a scalable real-world robotic learning system, addressing the associated challenges through a number of important design decisions, including a carefully designed multi-task data sharing strategy and multi-robot parallelized data collection. We train MT-Opt on a set of 12 real-world tasks that include data collected over the course of over 1.5 years, and provide an extensive evaluation (spanning over 500 real-robot-hours) of our method together with a number of baselines. We show that at such scale, our system can quickly acquire new tasks by taking advantage of prior tasks via shared representations, new data-sharing strategies and learned policies. Our results indicate that, by learning multiple related tasks simultaneously, not only can we increase the data-efficiency of learning each of them, but also solve more complex tasks than in a single-task setup. We present our multi-task system as well as examples of some of the tasks that it is capable of performing in Fig. 1.

Figure 1: A) Data collection. B) Training objects. C) Sample of training tasks. D) Sample of behaviorally and visually distinct tasks such as covering, chasing, alignment, which we show our method can adapt to. MT-Opt learns new tasks faster (potentially zero-shot if there is sufficient overlap with existing tasks), and with less data compared to learning the new task in isolation.

## 2   Related Work

Multi-task learning, inspired by the ability of humans to transfer knowledge between different tasks [14], is a promising approach for sharing structure and data between tasks to improve overall efficiency. Multi-task architectures have been successful across multiple domains; natural language processing [15, 16, 17, 18] and computer vision [19, 20, 21, 22, 23, 24]. In this work, we apply multi-task learning concept in a RL setting to real robotic tasks.

Combining multiple task policies has been explored in RL by using gating networks [25, 26], conditioning policies on tasks [27], mapping tasks to parameters of a policy [28, 29, 30], distilling separate task policies into a shared multi-task policy [31, 32, 33, 34, 35, 36]. Advantages of multi-task learning for visual representations has been explored in [37]. Pinto and Gupta [4] use a shared multi-task neural network architecture, which is trained with separate task-specific losses. In contrast, in our work, we concentrate on tasks with a common loss structure within a Q-learning framework. Several works explore how to mitigate multi-task interference and conflicting objectives [38, 39]. In our experiments, we find that better data routing helps with not only better mitigating conflicting objectives but also improving learning efficiency through data reuse.

Learning complex skills has been addressed through hierarchical reinforcement learning with options [40, 41, 42], combining multiple sub-tasks [43, 44, 45, 46, 47, 48], reusing samples between tasks [49], relabeling experience in hindsight [50], introducing demonstrations [51, 52, 53, 54, 55, 56, 57]. A range of works employ autonomous supervision to learn diverse skills, e.g. by scaling up data collection [58], sampling suitable tasks [59] or goals [60] to practice, learning a task embedding space amenable to sampling [7], or learning a dynamics model and using model-predictive control to achieve goals [61, 62, 63, 64]. Riedmiller et al. [5] learn sparse-reward tasks by solving easier auxiliary tasks and reusing that experience for offline learning of more complex tasks. In Cabi et al. [65], previously collected experience is relabeled with new reward functions in order to solve new tasks without re-collecting the data. In our work, we similarly design techniques for reusing experience between related tasks and apply them to large-scale data collection on real robots.

# 3   MT-Opt: a Scalable Multi-Task RL System

Fig. 2 overviews our multi-task learning system. We devise a distributed, off-policy multi-task RL algorithm together with a multi-task visual success detector ($SD$) to learn multiple robotic manipulation tasks simultaneously. The $SD$, trained from video examples of desired outcomes, (Fig. 2A) in part determines how episodes are leveraged to train an RL policy (Fig. 2C). During evaluation and fine-tuning (Fig. 2B), at each time step, a policy takes as input a camera image and a one-hot encoding of the task, and sends a motor command to the robot. At episode end, the outcome image of this process is graded by the $SD$ that determines which tasks were accomplished successfully and assigns a sparse reward 0 or 1 for each task. The system then decides whether another task should be attempted or if the environment should be reset. We train this system with multiple robots, where each robot concurrently collects data for a different, randomly selected task. The generated episodes are used as offline data for training future policies (Fig. 2C) and are available to improve $SD$s.

In order to enable our system to take full advantage of multi-task training, we develop a number of components and evaluate their importance for overall performance. First, we use a single, multi-task deep neural network to learn a policy for all the tasks simultaneously, which

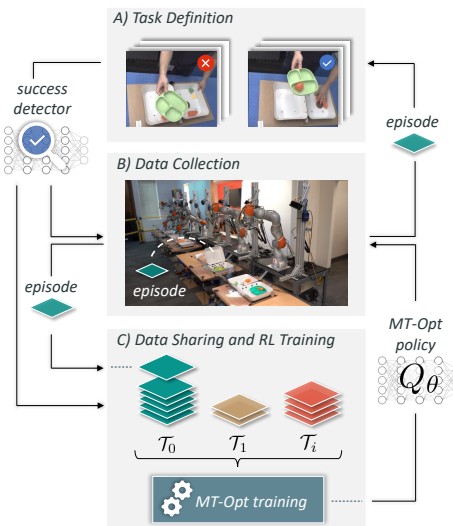

Figure 2: MT-Opt overview. A) The user defines a success detector ($SD$) for tasks through examples of desired outcomes, and relabeling outcomes of prior episodes. B) Utilizing the $SD$ and the MT-Opt policy, new episodes are collected for multiple tasks. C) Offline episodes enter the data-sharing pipeline that expands and re-balances the data used to train the MT-Opt policy, while optionally more on-policy data is collected, esp. for new tasks. This is an iterative process, which results in additional experiences that can be leveraged to define new tasks and train future policies.

enables parameter sharing between tasks. Second, we devise data management strategies that share and re-balance data across certain tasks. Third, since all tasks share data and parameters, we use some tasks as exploration policies for others, which aids in exploration. Although the individual components are based on previously studied concepts in multi-task learning, we show through our experiments that the particular choice and design of these components is essential for our large-scale multi-task learning system to effectively leverage data from all tasks to improve performance, particularly for "underrepresented" tasks that have the least available training data.

## 3.1   Multi-Task Reinforcement Learning Algorithm

We denote the multi-task RL policy as $\pi(\mathbf{a}|\mathbf{s}, \mathcal{T}_i)$, where $\mathbf{a} \in \mathcal{A}$ denotes the action, which in our case is the desired change in end-effector position, a change in yaw angle, binary gripper open and close commands and a termination command, $\mathbf{s} \in \mathcal{S}$ denotes the state, which corresponds to images from the robot's cameras, and $\mathcal{T}_i$ denotes an encoding of the $i^{\text{th}}$ task drawn from a categorical task distribution $\mathcal{T}_i \sim p(\mathcal{T})$, with $n$ possible categories, each corresponding to a different task. At each time step, the policy selects an action $\mathbf{a}$ given the current state $\mathbf{s}$ and the current task $\mathcal{T}_i$ set at the beginning of the episode, and receives a task-dependent reward $r_i(\mathbf{a}, \mathbf{s}, \mathcal{T}_i)$. The goal of the multi-task RL policy is to maximize the expected sum of rewards for all tasks drawn from the distribution $p(\mathcal{T})$. The episode finishes when the policy selects a TERMINATE action or reaches a step limit.

We build on the single-task QT-Opt algorithm [1], which itself is a variant of Q-learning [66], and learns a single-task optimal Q-Function by minimizing the Bellman error: $\mathcal{L}_i(\theta) = \mathbb{E}_{(\mathbf{s},\mathbf{a},\mathbf{s}') \sim p(\mathbf{s},\mathbf{a},\mathbf{s}')} \big[ D(Q_\theta(\mathbf{s}, \mathbf{a}), Q_T(\mathbf{s}, \mathbf{a}, \mathbf{s}')) \big]$, where $Q_T(\mathbf{s}, \mathbf{a}, \mathbf{s}') = r(\mathbf{s}, \mathbf{a}) + \gamma V(\mathbf{s}')$ is a target Q-value and $D$ is a divergence metric, such as cross-entropy, $\gamma$ is a discount factor, $V(\mathbf{s}')$ is the target value function of the next state computed using stochastic optimization of the form $V(\mathbf{s}') = \max_{\mathbf{a}'} Q(\mathbf{s}', \mathbf{a}')$, and the expectation is taken w.r.t. previously seen transitions $p(\mathbf{s}, \mathbf{a}, \mathbf{s}')$. Similarly to [1], we use the cross-entropy method (CEM) to perform the stochastic optimization to compute the target value function.

To extend this approach to the multi-task setting, let $\mathbf{s}^{(i)}, \mathbf{a}^{(i)}, \mathbf{s}'^{(i)}$ denote a transition that was generated by an episode $e^{(i)}$ for the $i^{\text{th}}$ task $\mathcal{T}_i$. Each transition could be used for multiple tasks. In the multi-task case, the loss becomes (with $(\mathbf{s}^{(i)}, \mathbf{a}^{(i)}, \mathbf{s}'^{(i)})$ being transitions generated by tasks $\mathcal{T}_i$):

$$\mathcal{L}_{\text{multi}}(\theta) = \mathbb{E}_{\mathcal{T}_i \sim p(\mathcal{T})} \left[ \mathcal{L}_i(\theta) \right] = \mathbb{E}_{\mathcal{T}_i \sim p(\mathcal{T})} \Big[ \tag{1}$$
$$\mathbb{E}_{p(\mathbf{s}^{(i)}, \mathbf{a}^{(i)}, \mathbf{s}'^{(i)})} \Big[ D(Q_\theta(\mathbf{s}^{(i)}, \mathbf{a}^{(i)}, \mathcal{T}_i), Q_T(\mathbf{s}^{(i)}, \mathbf{a}^{(i)}, \mathbf{s}'^{(i)}, \mathcal{T}_i)) \Big] \Big].$$

## 3.2 Sharing Data Between Tasks

One advantage of an off-policy RL algorithm is that collected experience can be used to update the policy for other tasks, not just the task for which it was originally collected. This section describes how we effectively train with multi-task data through *task impersonation* and *data re-balancing*, as summarized in Fig. 3.

We leverage experience sharing at the episode level rather than the individual transition level. The goal is to use all transitions of an episode $e^{(i)}$ generated by task $\mathcal{T}_i$ to aid in training a policy for a *set of* $k_i$ tasks $\mathcal{T}_{\{k_i\}}$. We refer to this process as *task impersonation* (see Algorithm 1), where the impersonation function $f_I$ transforms episode data collected for one task into a set of episodes that can be used to also train other tasks, i.e.: $e^{\{k_i\}} = f_I(e^{(i)})$. Note that in general case $\{k_i\}$ is a *subset* of all tasks $\{n\}$, and it depends on the original task $\mathcal{T}_i$ that the episode $e^{(i)}$ was collected for.

We consider two ends of the data-sharing spectrum: an identity impersonation function $f_{I_{\text{orig}}}(e^{(i)}) = e^{(i)}$, where no task impersonation

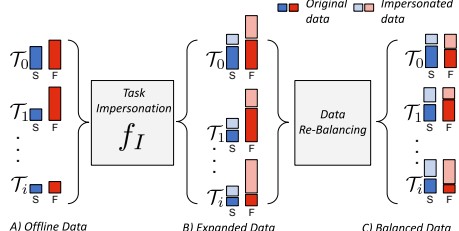

Figure 3: Task impersonation: episodes are routed to train relevant tasks, and data re-balancing where the ratio of success (S) and failure (F) episodes and proportion of data per task is controlled. Pale blue and pale red show additional training data coming from other tasks. The bar height indicates different amount of data across tasks and successful outcomes.

---

**Algorithm 1** Task Impersonation

---

**procedure** $f_I(e^i : \text{original\_episode})$
    expanded\_episodes = []
    $SD\{k_i\} \leftarrow$ set of SDs relevant to task $T_i$
    **for** $SD_k$ in $SD\{k_i\}$ **do**
        // $e^k$: $e^i$ but rewards for task $T_k$ not $T_i$
        $e^k = SD_k(e^i)$
        expanded\_episodes.append($e^k$)
    **return** expanded\_episodes

---

takes place, i.e. an episode $e^{(i)}$ generated by task $\mathcal{T}_i$ is used to train the policy only for that task; and $f_{I_{\text{all}}} = e^{\{n\}}$, where each task shares data with *all* $n - 1$ tasks (maximal data sharing). While $f_{I_{\text{orig}}}$ fails to leverage the reusable nature of multi-task data, $f_{I_{\text{all}}}$ results in many unrelated episodes used as negative examples for the target task.

For MT-Opt, we devise a new task impersonation strategy $f_{I_{\text{skill}}}$ that uses more fine-grained similarities between tasks. We refer to it as a skill-based task-impersonation strategy, where we overload the term "skill" as a set of tasks that share semantics and dynamics, yet can start from different initial conditions or operate on different objects. In this work we manually assign each task to a particular skill based on the subjective similarity of the required manipulation primitive. For example, tasks such as *place-object-on-plate* and *place-object-in-bowl* belong to the same skill. Our impersonation function $f_{I_{\text{skill}}}$ allows us to impersonate an episode $e^{(i)}$ only as the tasks belonging to the same skill as $\mathcal{T}_i$. Namely, given an episode $e_i$ generated by task $\mathcal{T}_i$, a skill $S_j$ that task belongs to is detected. The $e_i$ will be impersonated only for the tasks $\mathcal{T}_{\{S_j\}}$ belonging to that particular skill. In order to avoid impersonating too many episodes and overloading the replay buffer, we introduce a stochastic impersonation function. An impersonated episode candidate will be routed to training with the probability $p_s$ if it's a success, or with probability $p_f$ if it's a failure. We provide more details on the specific parameters in the Appendix 7.3. In our experiments, we conduct ablation studies comparing $f_{I_{\text{skill}}}$ (ours) with other task impersonation strategies.

While training, due to the design of our task impersonation mechanism, as well as the variability in difficulty between tasks, the resulting training data stream often becomes highly imbalanced, see Fig. 3B. To address this, we re-balance each batch both between tasks, such that the relative proportion of training data for each task is equal, and within each task, such that the relative proportion of successful and unsuccessful examples is kept constant.

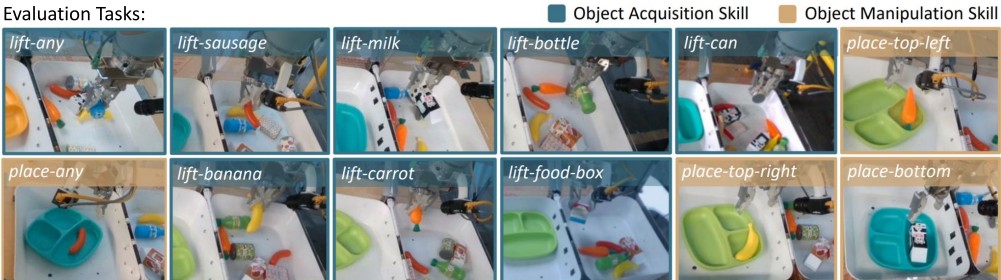

Evaluation Tasks:    ■ Object Acquisition Skill    ■ Object Manipulation Skill

*lift-any*   *lift-sausage*   *lift-milk*   *lift-bottle*   *lift-can*   *place-top-left*

*place-any*   *lift-banana*   *lift-carrot*   *lift-food-box*   *place-top-right*   *place-bottom*

Figure 5: 12 evaluation tasks, giving rise to Object Acquisition and Object Manipulation skills.

## 4   SD-Based Rewards and Continuous Data Collection

We learn tasks evaluated on the final state of an episode. This sparse-reward assumption allows us to train a multi-view multi-task CNN success detector model ($SD$), conditioned on task ID, which given 3 final images from 3 different camera angles, infers task success (see Appendix 7.1 for details).

To generate training data for the $SD$, we ask users to collect positive and negative examples of positive and negative outcomes of a task. These examples are not demonstrations – just examples of what successful completion (i.e., the final state) looks like. We use this data to train the initial version of the multi-task $SD$, which we then continuously fine-tune given the images gathered during autonomous data collection to avoid out of distribution images that might be caused by various real-world factors such as different lighting conditions, and novel states which the robot discovers. We continue to manually label such images and incrementally retrain $SD$ to obtain the most up-to-date $SD$. In result, we label $\approx 5,000$ images per task and provide more details in the Appendix 8.

Our main observation w.r.t. the multi-task data collection is the use of easier tasks to bootstrap learning of more complex tasks. In particular, an average MT-Opt policy for simple tasks might occasionally yield episodes successful for harder tasks. Over time, this allows us to start training an MT-Opt policy now for the harder tasks, and consequently, to collect better data for those tasks. To kick-start this process and bootstrap our two simplest tasks, we use two crude scripted policies for picking and placing (see Sec. 7.2 for details) following prior work [1]. To simplify exploration for longer-horizon tasks, we allow the individual tasks to be ordered sequentially, where one task is executed after another using simple heuristics such as executing placing tasks after lifting tasks. Our dataset grows over time w.r.t. the amount of per-task data as well as percentage of successful episodes for all the tasks. All data

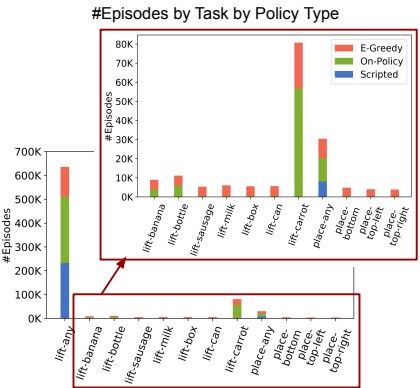

Figure 4: Offline dataset properties. Our data collection strategy collects data for multiple tasks, where it uses easier, more general tasks (e.g. *lift-any*) to bootstrap learning of more specialized tasks (e.g. *lift-carrot*). The resulting dataset is imbalanced across the distribution of exploration policies per task and success rate per task.

from a variety of different policies is accumulated over time: different versions of trained policies, their epsilon-greedy counterparts ($\epsilon = 20\%$), and scripted policies. Importantly, this data collection process results in an imbalanced dataset, as shown on Fig. 4. Our data impersonation and re-balancing methods address this imbalance by efficiently expanding and normalizing data.

## 5   Experiments

The goal of our real-world experiments is to answer the following questions: **(1)** How does MT-Opt perform, quantitatively and qualitatively, on a large set of vision-based robotic manipulation tasks? **(2)** Does training a shared model on many tasks improve MT-Opt's performance? **(3)** Does data sharing improve performance of the system? **(4)** Can we use easier tasks to bootstrap learning of more difficult tasks? **(5)** Can MT-Opt quickly learn distinct new tasks by adapting learned skills?

## 5.1 Experimental Setup

MT-Opt provides a general robotic skill learning framework that we use to learn multiple tasks, including semantic picking (i.e., picking an object from a particular category), placing into various fixtures (e.g., placing a food item onto a plate). We focus on basic manipulation tasks that require repositioning objects relative to each other. A wide range of manipulation behaviors fall into this category, from simple bin-picking to more complex behaviors, such as covering items with a cloth. As described in Sec. 4, our task set grows over time allowing for more and more tasks to be similar. To analyze and ablate various properties of MT-Opt, we focus on a set of different but similar tasks, where multi-task learning shows maximal effectiveness. In the following experiments, we use a set of 12 tasks for quantitative evaluation of our algorithm. These 12 tasks include a set of plastic food objects and divided plate fixtures and they can be split into "object acquisition" and "object manipulation" skills. Since the criteria of what constitutes a task are not well defined, in our system, different tasks correspond to different success conditions as defined by our $SD$. Our most general object acquisition task is *lift-any*, where the goal is to singulate and lift any object. We also define 7 semantic lifting tasks, where the goal is to search for and lift a particular object (ex: carrot). Importantly, these semantic tasks require search in clutter and singulation before the actual picking can be performed. The placing tasks utilize a divided plate where the simplest task is to place the previously lifted object anywhere on the plate (*place-any*). Harder tasks require placing the object into a particular section of a divided plate, oriented arbitrarily. Fig. 5 visualizes the tasks.

All polices are trained with offline RL from a large dataset summarized in Fig. 4. The resulting policy is deployed on 7 robots attempting each task 100 times for evaluation. To reduce the variance of the evaluation, we shuffle the bins after each episode and use a standard evaluation scene (see Appendix, Fig. 17), where all of objects are present, hence all 12 evaluation tasks are feasible.

## 5.2 Quantitative and Qualitative Evaluation of MT-Opt

Fig. 6 shows MT-Opt success rates on the 12 evaluation tasks. We compare to three baselines: (i) single-task QT-Opt [1], where each per-task policy is trained separately using only data collected specifically for that task, (ii) an enhanced QT-Opt baseline, which we call QT-Opt Multi-Task, where we train a shared policy for all the tasks but there is no data impersonation or re-balancing between the tasks, and (iii) a Data-Sharing Multi-Task baseline that is based on the data-sharing strategy presented in [65], where we also train a single Q-Function but the data is shared across all tasks. From the average performance across all task, we observe that MT-Opt outperforms the baselines, in some cases with $\approx 3\times$ average improvement. While the single task QT-Opt baseline performs similarly to MT-Opt for the task where we have the most data (see data statistics in Fig. 4), *lift-any*, its performance drastically drops (to $\approx 1\%$) for underrepresented tasks, such as *lift-can*. Note, that we are not able run this baseline for the placing tasks, since they require a separate task to lift the object, which is not present in the single-task baseline. A similar observation applies to QT-Opt Multi-Task, where the performance of rare tasks increases compared to QT-Opt, but is still $\approx 4\times$ worse on average than MT-Opt. Sharing data across all tasks also results in a low performance for semantic lifting and placing tasks and, additionally, it appears to harm the performance of the indiscriminate lifting and placing tasks. The MT-Opt policy, besides attaining the 89% success rate on (*lift-any*), also performs the 7 semantic lifting tasks and the 4 placing and rearrangement tasks at a significantly higher success rate than all baselines. Since all ablated experiments are trained on the same exact offline dataset (except for the case where the multiple policies are trained on the per-task shard of the dataset) and have the same computation budget, we explain these performance gaps by the way MT-Opt shares the representations and data, and provide a more comprehensive analysis of these factors in the following experiments. To highlight learned behaviors, Fig. 7 shows the policy work the carrot out of the corner before picking it up.

| Parameter Sharing (Success Rate) | | |
|---|---|---|
| Model: | 2-Task | 12-Task |
| *lift-any* | 0.82 | **0.89** |
| *place-any* | 0.63 | **0.85** |

Table 1: Parameter sharing: the policy that learns two tasks (*lift-any*, *place any*) in addition to 10 other tasks outperforms a policy trained only for the two target tasks. Both policies are trained on the same offline dataset.

Fig. 4, shows amounts and type of data collected for various tasks. Tasks such as *lift-carrot* and *lift-bottle*, which have more data, especially on-policy data, have higher success rates than underrepresented tasks, such as *lift-box*. The performance of these underrepresented tasks could be further improved by focusing the data collection on these tasks.

## 5.3 Sharing Representations Between Tasks

To explore the benefits of training a single policy on multiple tasks, we compare the 12-task MT-Opt policy with a 2-task policy that learns *lift-any* and *place-any*. Both of these policies are evaluated on (*lift-any* and *place-any*). We use the same $f_{I_{skill}}$ task impersonation strategy, and the same offline dataset (i.e. both policies use the data from the extra 10 narrower tasks, which is impersonated as *lift-any* and *place-any* data) without on-policy fine-tuning, so data-wise the experiments are identical.

The 12-task policy outperforms the 2-task policy *even on the two tasks that the 2-task policy is trained on* (Table 1), suggesting that training multiple tasks not only enables the 12-task policy to perform more tasks, but also improves its performance on the tasks through sharing representations. This may suggest that the additional supervision provided by training on more tasks has a beneficial effect on the shared representations.

## 5.4 Data Sharing Between Tasks

To test the influence of data-sharing and re-balancing on the multi-task policy's performance, we compare our task impersonation strategy $f_{I_{skill}}$ introduced in Sec. 3.2 to a baseline impersonation function that does not share the data between the tasks $f_{I_{orig}}$, and a baseline where each task is impersonated for all other tasks $f_{I_{all}}$. In our skill-based task impersonation strategy $f_{I_{skill}}$, the data is expanded only for the class of tasks having similar visuals, dynamics and goals. We also test data re-balancing strategy, where we re-balance each training batch between the tasks as well as within each task to keep the relative proportion of successful and unsuccessful trials the same.

The results of this experiment are in Table 2, with the full results reported in the Appendix, Table 4. Sharing data among tasks using our method of task impersonation and re-balancing provides significant improvement across all the tasks, with improvements of up to $10x$ for some tasks. The full data-sharing strategy performs worse than both the no-data-sharing baseline and our method, suggesting that

| Data Strategies (min, mean, max, mean of low data tasks) | | |
|---|---|---|
| Imp. | Data Re-Balancing Strategy | |
| Fn | uniform sampling | task re-balanced sampling |
| $f_{I_{orig}}$ | 0.10 / 0.32 / 0.94 / 0.18 | 0.16 / 0.55 / 0.85 / 0.42 |
| $f_{I_{all}}$ | 0.07 / 0.21 / 0.62 / 0.13 | 0.02 / 0.35 / **0.95** / 0.21 |
| $f_{I_{skill}}$ | 0.17 / 0.46 / 0.88 / 0.32 | **0.29** / **0.58** / 0.89 / **0.50** Ours |

Table 2: Min, average and max task performance across 12 tasks, as well as average performance across 6 tasks having least data ($\approx 6K$ episodes) for different data-sharing strategies. $f_{I_{skill}}$ impersonation and data re-balancing are complimentary: they both improve over the baselines, while the best effect is achieved by combining both. The effect is especially pronounced for the underrepresented tasks.

*naïvely sharing all data across all tasks is not effective*. Because of our data-collection strategy, the resulting multi-task dataset contains much more data for broader tasks (e.g., *lift-any*) than for more narrow, harder tasks (e.g., *lift-box*), as shown in Fig. 4. Without any additional data-sharing and re-balancing, this data imbalance causes the baseline strategy $f_{I_{orig}}$ to attain good performance for the easier, overrepresented tasks, but poor performance on the harder, underrepresented tasks (see Table 2, first row), whereas our method performs substantially better on these tasks.

## 5.5 Using Easier Tasks to Bootstrap Harder Tasks

To explore question (4), we study whether learning an easier, broader task (*lift-any*) helps with a structurally related task that is harder, more specific (*lift-sausage*). We separate out data for *lift-sausage* which consists of $5400$ (nearest hundred) episodes collected for that task (i.e. $4600$ failures and $800$ successes). In addition, there are $11200$ episodes of successful sausage lifting and as many as $740K$ failures that were collected during the *lift-any* task. Combining the *lift-sausage* data and the extra successes from *lift-any* yields $16600$ episodes ($12000$ successes and $4600$ failures). To investigate the influence of MT-Opt and task impersonation on the bootstrap problem, we compare our 12-task MT-Opt policy to a single-task policy trained on these $16600$ episodes, including the same set of successful *lift-sausage* episodes as MT-Opt, but not including failures from other tasks.

The single-task policy learned from the $16600$ episodes yields performance of $3\%$. MT-Opt, using impersonated successes and failures, achieves $39\%$ success for the same task, a $\approx 10\times$ improvement. Both experiments use identical data representing successful episodes. The benefits of MT-Opt are twofold. First, we leverage an easier *lift-any* task to collect data for the harder *lift-sausage* task. Second, MT-Opt benefits from the additional failures impersonated from other tasks. These failures, which often include successful grasps of non-target objects, when re-balanced as described in Sec. 3.2, results in the significant task performance boost. This demonstrates the value of both successful and unsuccessful data collected by other tasks for learning new tasks.

### 5.6 Learning New Tasks with MT-Opt

In addition to retroactive relabelling of prior data, MT-Opt can learn new tasks via *proactive* adaptation of known tasks, even ones that are visually and behaviorally different than the initial training set. To accommodate future tasks, we start learning with a larger one-hot vector. Once a new task is defined and its success detector is trained, we allocate a next available one-hot task ID to this task. To demonstrate this, we perform a fine-tuning experiment, bootstrapping from the MT-Opt 12-task policy (Sec. 5.2). We use the MT-Opt policy to collect data for unseen tasks: *lift-cloth* and *cover-object* ( Fig. 7 bottom row). Unlike *lift-sausage*, prior to starting collection of these new tasks, no episodes in our offline dataset can be relabelled as successes for these tasks.

We follow the continuous data collection process (Sec. 4): we define and train the success detector for the new tasks, collect initial data using our *lift-any* and a *place-any* policies, and fine-tune a 14-task MT-Opt model that includes all prior as well as the newly defined tasks. While the new tasks are visually and semantically different, in practice the above mentioned policies give reasonable success rate necessary to start the fine-tuning. We switch to running the new policies on the robots once they are at parity with the *lift-any* and *place-any* policies. After $11K$ *pick-cloth* attempts and $3K$ *cover-object* attempts (requiring $< 1$ day of data collection on 7 robots), we obtain an extended 14-task MT policy that performs cloth picking at $70\%$ success and object covering at $44\%$ success. The policy trained only for these two tasks, without support of our offline dataset, yields performance of $33\%$ and $5\%$ respectively, confirming the hypothesis that MT-Opt method is beneficial even if the target tasks are sufficiently different, and the target data is scarce. By collecting additional $10K$ *pick-cloth* episodes and $6K$ *cover-object* episodes, we further increase the performance of 14-task MT-Opt to $92\%$ and $79\%$, for cloth picking and object covering respectively. We perform this fine-tuning procedure with other novel tasks such as previously unseen transparent bottle grasping, which reaches a performance of $60\%$ after 4 days of collection.

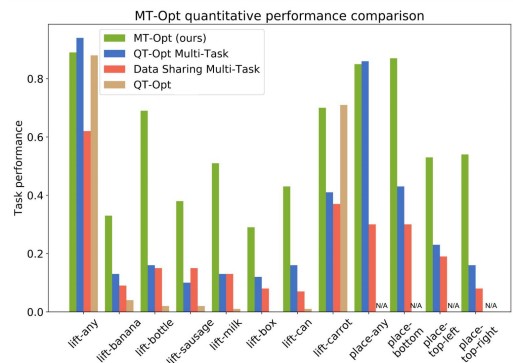

Figure 6: Quantitative evaluation of MT-Opt across 12 tasks. QT-Opt [1] trains each task individually using only data collected for that task. QT-Opt Multi-Task [1] trains a single network for all tasks but does not share the data between them. Data-Sharing Multi-Task also trains a single network for all tasks and shares the data across all tasks without further re-balancing. MT-Opt (ours) provides a significant improvement over the baselines, especially for the harder tasks with less data.

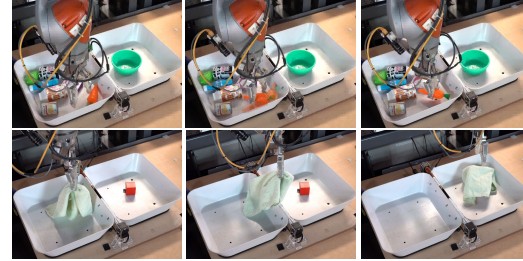

Figure 7: Top: Example of *pick-carrot*. The robot repositions the carrot out of the corner to pick it. Bottom: *cover-object*. The deformable covers the object.

### 5.7 Limitations of MT-Opt

While we show successful execution of MT-Opt on a number of tasks and its potential to scale, it is important to list limitations of this system. i) The shown tasks are short-horizon ($\leq 20$ steps). Long-horizon real-world robotic tasks are still hard for RL methods; ii) The task outcome is determined from the final episode frame. In theory, this could be extended to learned intermediate rewards; iii) Our tasks leverage an automated reset mechanism. While this is not an assumption of the method, we have not demonstrated learning any reset-free behaviors; iv) Learning drastically different skills is challenging, as our method works best when adapting more similar tasks. It remains an open question as to how far MT-Opt can be scaled to a very diverse set of skills; v) MT-Opt is 4DoF with gripper and terminate actions. Learning higher DoF tasks is challenging yet principally possible.

## 6 Conclusion

We presented a general multi-task learning framework, MT-Opt, that encompasses a multi-task data collection system, a scalable success detector framework, and a deep RL method that is able to effectively utilize the multi-task data. Our experiments show that by sharing data and parameters across tasks we significantly increase the data efficiency of learning individual tasks and that MT-Opt quickly acquires new tasks via the shared multi-task representations and exploration strategies.

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

# 7 Appendix

## 7.1 Neural Network Architecture

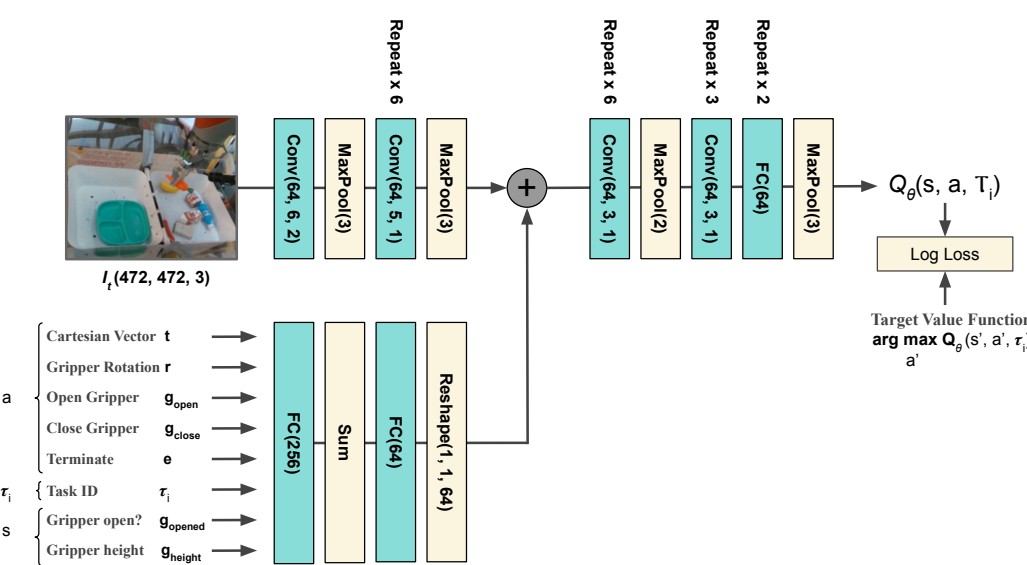

Figure 8: The architecture of MT-Opt Q-function. The input image is processed by a stack of convolutional layers. Action vector, state vector and one-hot vector $\mathcal{T}_i$ representing the task of interest are processed by several fully connected layers, tiled over the width and height dimension of the convolutional map, and added to it. The resulting convolutional map is further processed by a number of convolutional layers and fully connected layers. The output is gated through a sigmoid, such that Q-values are always in the range [0, 1].

We model the Q-function for multiple tasks as a large deep neural network whose architecture is shown in Fig. 8. This network resembles one from [1]. The network takes the monocular RGB image part of the state $s$ as input, and processes it with 7 convolutional layers. The actions $a$ and additional state features ($g_{status}$, $g_{height}$) and task ID $\mathcal{T}_i$ are transformed with fully-connected layers, then merged with visual features by broadcasted element-wise addition. After fusing state and action representations, the Q-value $Q_\theta(s, a)$ is modeled by 9 more convolutional layers followed by two fully-connected layers. In our system the robot can execute multiple tasks from in the given environment. Hence the input image is not sufficient to deduce which task the robot is commanded to execute. To address that, we feed one-hot vector representing task ID into the network to condition Q-Function to learn task-specific control.

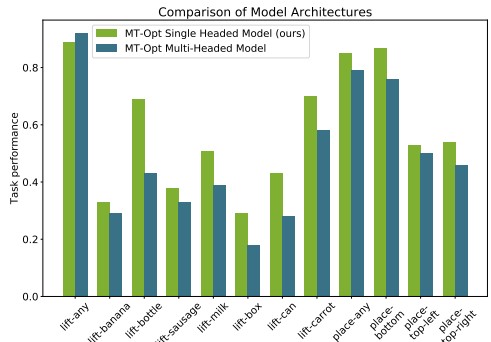

Figure 9: Comparison of single-headed and multi-headed neural networks approximating the Q-function. In both cased task ID was fed as the input to the network. Multi-headed architecture of the Q-function under-performs on a wide range of tasks, winning only on *lift-any* tasks which has most of the data.

In addition to feeding task ID we have experimented with multi-headed architecture, where $n$ separate heads each having 3 fully connected layers representing $n$ tasks were formed at the output of the network. Fig.9 shows that performance of the system with the multi-headed Q-function architecture is worse almost for all tasks. We hypothesize that dedicated per task heads "over-compartmentalizes" task policy, making it harder to leverage shared cross-task representations.

## 7.2 Description of Scripted Policies

As discussed in Section 4 we use two crude scripted policies to bootstrap easy generic tasks.

**Scripted Picking Policy:** To create successful picking episodes, the arm would begin the episode in a random location above the right bin containing objects. Executing a crude, scripted policy, the arm is programmed to move down to the bottom of the bin, close the gripper, and lift. While the success rate of this policy is very low ($\approx 10\%$), especially with the additional random noise injected into actions, this is enough to bootstrap our learning process.

**Scripted Placing Policy:** The scripted policy programmed to perform placing would move the arm to a random location above the left bin that contains a fixture. The arm is then programmed to descend, open the gripper to release the object and retract. This crude policy yields a success rate of (47%) at the task of placing on a fixture (plate), as the initial fixture is rather large. Data collected by such a simplistic policy is sufficient to bootstrap learning.

## 7.3 $f_{I_{skill}}$ impersonation strategy details

Task impersonation is an important component of the MT-Opt method. Given an episode and a task definition, the $SD$ classifies if that episode is an example of a successful task execution according to that particular goal definition. Importantly, both the success and the failure examples are efficiently utilized by our algorithm. The success example determines what the task is, while the failure example determines what the task is *not* (thus still implicitly providing the boundary of the task), even if it's an example of a success for some other task. Fig.11 shows offline success rates and Fig.12 shows by how much the per task data is expanded using the $f_{I_{\text{skill}}}$ impersonation function.

In Section 3.2 we discuss a problem arising when using a naive $f_{I_{all}}$ episodes impersonation function, and suggest a solution to impersonate data only within the boundaries of a skill. Namely, given an episode $e_i$ generated by task $\mathcal{T}_i$, a skill $S_j$ that task belongs to is detected. The $e_i$ will be impersonated only for the tasks $\mathcal{T}_{\{S_j\}}$ belonging to that particular skill.

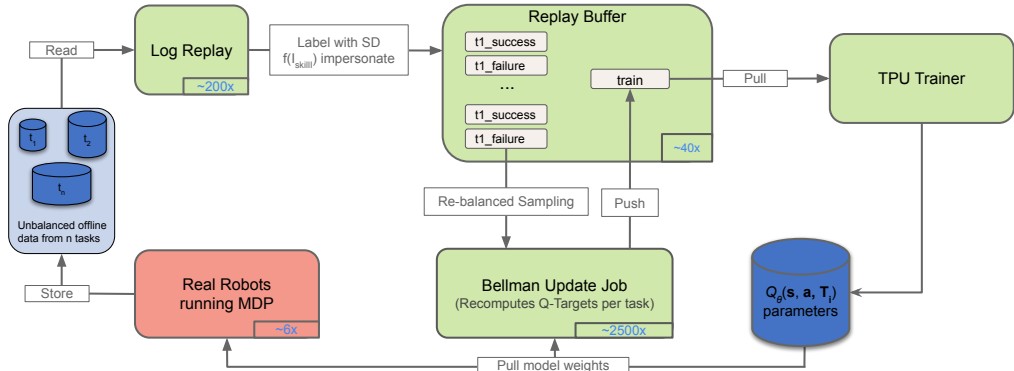

Figure 10: **System overview:** Task episodes from disk are continuously loaded by LogReplay job into task replay buffers. LogReplay process assigns binary reward signal to episodes using available Success Detectors and impersonates episodes using $f_{I_{skill}}$ (or other strategy). Impersonated episodes are compartmentalized into dedicated per task buffers, further split into successful and failure groups. Bellman Update process samples tasks using re-balancing strategy to ensure per task training data balancing and computes Q-targets for individual transitions, which are placed into train buffer. These transitions $(\mathbf{s}, \mathbf{a}, \mathcal{T}_i)$ are sampled by the train workers to update the model weights. The robot fleet and Bellman Update jobs are reloading the most up to date model weights frequently.

Note, that sometimes impersonation for all $\mathcal{T}_{\{S_j\}}$ tasks within a skill could result in too excessive data sharing. For example, the bulk of the data for our *object-acquisition* skill represents variants of tasks involving foods objects. If we want to learn a new task within the same skill using visually significantly different objects, e.g. transparent bottles, all offline episodes involving the plastic objects will be (correctly) impersonated as failures for the *lift-transparent-bottle* task. That is, a few intrinsic failures for that task will be diluted in large set of artificially created negatives.

| Primary SD Name | Total Count | Success Count | Failure Count | Success Rate | F. Neg. Rate | F. Pos. Rate | Other F. Neg. Rate | Other F. Pos. Rate |
|---|---|---|---|---|---|---|---|---|
| *lift-any* | 16064 | 7395 | 8672 | 46% | 1% | 2% | 0% | 0% |
| *lift-banana* | 6255 | 510 | 5745 | 8% | 2% | 1% | 0% | 1% |
| *lift-bottle* | 6472 | 430 | 6042 | 7% | 5% | 1% | 0% | 1% |
| *lift-sausage* | 6472 | 461 | 6011 | 7% | 3% | 0% | 0% | 1% |
| *lift-milk* | 6472 | 158 | 6314 | 2% | 7% | 0% | 3% | 9% |
| *lift-box* | 6467 | 487 | 5980 | 8% | 1% | 1% | 0% | 2% |
| *lift-can* | 6467 | 270 | 6197 | 4% | 2% | 0% | 3% | 3% |
| *lift-carrot* | 6481 | 911 | 5570 | 14% | 0% | 1% | 0% | 0% |
| *place-any* | 3087 | 1363 | 1724 | 44% | 1% | 2% | 0% | 0% |
| *place-bottom* | 2893 | 693 | 2200 | 24% | 2% | 1% | 1% | 3% |
| *place-top-left* | 2895 | 346 | 2549 | 12% | 10% | 0% | 3% | 8% |
| *place-top-right* | 2897 | 312 | 2585 | 11% | 4% | 0% | 0% | 5% |

Table 3: Success detection holdout data statistics. Table shows success detector error rate for held out labelled success detector data. We split out the evaluation dataset based on the robot, e.g. all data generated by Robot #1 is used for evaluations and not for training. This strategy results in a much better test of generalization power of the success detector, compared to the conventional way to split out 20% of the data randomly for evaluation. The Other Task False [Positive/Negative] Rates columns indicates how well the success detector for a task A classifies outcomes for all other tasks. For example we want to ensure that a successful *lift-carrot* episode does not trigger *lift-banana* success, i.e. not only a success detector should manifest its dedicated task success, but also reliably reason about other related tasks. This "contrastiveness" property of the success detectors is of great importance in our system. As success detectors determine tasks data routing and experience sharing, an error in this tasks data assignment would drive anti-correlated examples for each task, resulting in a poor performance of the system.

To solve this issue we introduce a stochastic impersonation function. An impersonated episode candidate will be routed to training with the probability $p_s$ if it's a success, or with probability $p_f$ if it's a failure. We experiment with $p_s = 1.0$, and $p_f <= 1.0$. The reasoning is that it's always desirable to utilize surplus impersonated examples of a *successful* task execution, but it could be better to utilize only a fraction of the surplus *failures* to balance intrinsic v.s. artificial failures for that task.

This gives rise to the $f_{I_{skill}}(p_s, p_f)$ impersonation function which is suitable in some situations explained above.

### 7.4 Distributed Asynchronous System

Fig.10 provides an overview of our large scale distributed Multi-Task Reinforcement Learning system.

# 8 Reward Specification with Multi-Task Success Detector

Training a visual success detector is an iterative process, as a new task initially has no data to train from. We have two strategies to efficiently create an initial $SD$ training dataset for a new task. 1) We collect 5Hz videos from 3 different camera angles where every frame of the video a human is demonstrating task success, and then a short video demonstrating failure (see exam-

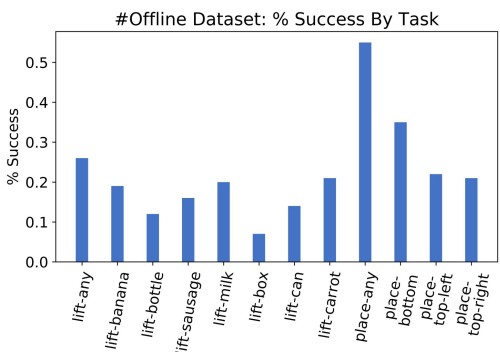

Figure 11: Effective success rate for each task in our offline dataset. This plot represents the distribution of successes within the entirety of our offline dataset collected over time from many policies, not the performance of any particular policy.

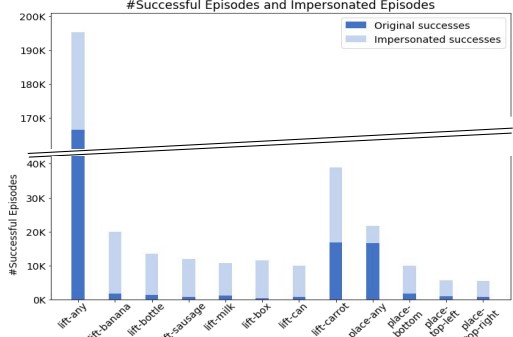

Figure 12: Practical effect of task impersonation for successful outcomes. Dark blue indicates data specifically collected for a task; light blue indicates episodes impersonated from some other tasks which happen to be a success for the target task.

Overhead Camera    Right Camera    Left Camera

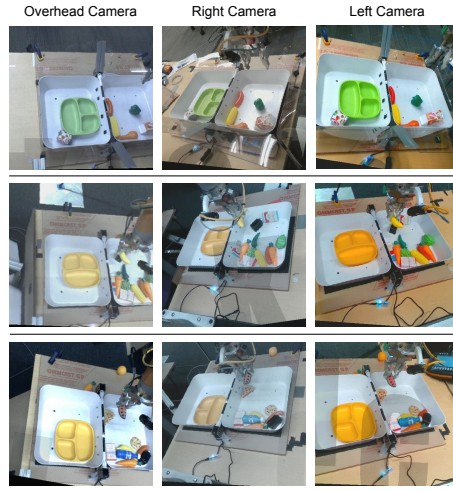

Figure 13: $SD$ training images. Each row represents a set of images captured at the same time that are fed into the $SD$ model. These images demonstrate our train-time $SD$ data augmentation process as they have been distorted via cropping, brightening, rotating, and superimposing of shadows.

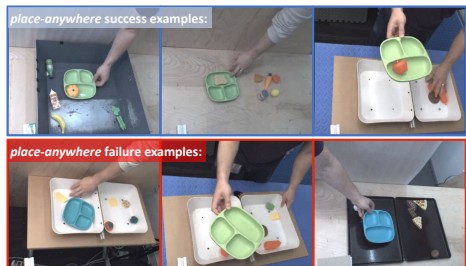

Figure 14: Video frames for the *place-anywhere* task. Success and failure videos are iteratively captured in pairs to mitigate correlations with spurious workspace features such as hands of the user, backgrounds, bins, and distractor objects.

ples in Fig. 14. Note that the user shows the desired and non-desired *outcome* of the task, not to be confused with demonstrations of how the task needs to be done.

The intention here is to de-correlate spurious parts of the scene from task-specifics. This process is repeated for approximately 30 minutes per task. 2) We relabel data from a policy that occasionally generated success for the new task (e.g., relabel *lift-any* data for *lift-carrot* task.).

The user would then change the lighting, switch out the objects and background, and then collect another pair of example videos (see Fig. 14 for example one video where there is always something on a plate being moved around paired with another video where there is never anything on a plate).

Once the initial $SD$ is trained, we can train an RL policy, and begin on-policy collection. We continue to label on-policy data which keeps coming for the new task until $SD$ is reliable. Table 3 shows false positive and false negative error rates on holdout data for the $SD$ model used in our ablations. Our holdout data consisted of all images from a particular robot.

During the $SD$ training process, the data is artificially augmented to improve generalization, which involves cropping, brightening, rotating, and superimposing random shadows onto the images.

Fig. 13 shows training images after these distortions have been applied. Our success detector model is trained using supervised learning, where we balance the data between success and failures as well as tasks. We use the architecture that is based on that from [67] with the exception of the action conditioning as it is not needed for this classification task. For each task the network outputs the probability representing whether a given state was a success or failure for the corresponding task. The model receives three images as an input that come from an over-the-shoulder camera (same image as RL policy), and two additional side cameras. These side camera images are only used by the $SD$ model, not the RL model. The additional cameras ensured that the task goals would be unambiguous, with a single camera, it was often difficult for a human to discern from an image whether or not the task had succeeded.

A breakdown of the labelled $SD$ training data is provided in Fig. 15. While training SD, we incorporated data sharing logic based on task feasibility. For example any success for *lift-carrot* would also be marked as failure for all other instance lifting tasks, and as a success for *lift-any*. In this manner, the original set of labelled data shown in Fig. 15 could act effectively as a much larger dataset for all tasks, where successes of one task often worked an interesting negatives for other

tasks. Additionally we balanced the proportion of success and failure examples per task seen by the model during training.

# 9   Robot Setup

In order for our system to be able to learn a vision-based RL policy that can accomplish multiple tasks, we need to collect a large, diverse, real-robot dataset that represents data for various tasks.

To achieve this goal, we set up an automated, multi-robot data collection system where each robot picks a task $\mathcal{T}_i$ to collect the data for. Collected episode is stored on disk along with the $\mathcal{T}_i$ bit of information. Our learning system can then use this episode collected $\mathcal{T}_i$ for to train a set of other tasks utilizing MT-Opt data impersonation algorithm. Once the episode is finished, our data collection system decides whether to continue with another task or perform an automated reset of the workspace.

In particular, we utilize 7 KUKA IIWA arms with two-finger grippers and 3 RGB cameras (left, right, and over the shoulder). In order to be able to automatically reset the environment, we create an actuated resettable bin, which further allows us to automate the data collection process. More precisely, the environment consists of two bins (with the right bin containing all the source objects and the left bin containing a plate fixture magnetically attached anywhere on the workbench) that are connected via a motorized hinge so that after an episode ends, the contents of the workbench can be automatically

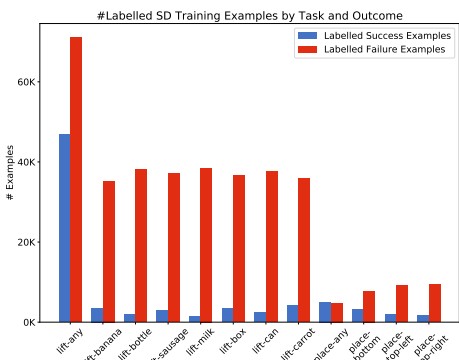

Figure 15: Counts of labelled $SD$ training data by task and outcome. This data was generated either from human video demonstration, or by labelling terminal images from episodes produced by a robot. Note, that not all of the negatives were hand-labelled. As we may know dependencies between the tasks, e.g. that a success for *lift-carrot* is always a failure for *lift-banana*, we can automatically generate negative examples. Similarly, all successes for the semantic lifting tasks are also successes for the *lift-any* task.

shuffled and then dumped back into the right bin to start the next episode. Fig. 16 depicts the physical setup for data collection and evaluation. This data collection process allows us to collect diverse data at scale: 24 hours per day, 7 days a week across multiple robots.

One episode has $\approx 10$ steps on average, taking $\approx 25$ seconds to be generated on a robot, including environment reset time. This accounts to $\approx 3300$ episodes/day collected on a single robot, or $\approx 23K$ episodes/day collected across our fleet of 7 robots.

## 9.1   Details of Data Collection to bootstrap a Multi-Task System

This section contains more details on the data collection process introduced in Section 9.1. Real world robot data is noisy. For this project nearly 800,000 episodes were collected through the course of 16 months. The data was collected over different:

1. Locations: Three different physical lab locations.
2. Time of day: Robots ran as close to 24x7 as we could enable.
3. Robots: 6-7 KUKAs with variations in background, lighting, and slight variation in camera pose.
4. Success Detectors: We iteratively improved our success detectors.
5. RL training regimes: We developed better training loops hyper-parameters and architectures as time went on.
6. Policies: Varied distribution of scripted, epsilon greedy, and on-policy data collection over time.

Our data collection started in an original physical lab location, was paused due to COVID-19, and the robots were later setup at a different physical lab location affecting lighting and backgrounds.

| Task Name | #Eps. | QT-Opt | $f_{I_{\text{orig}}}$, rand QT-Opt MultiTask | $f_{I_{\text{orig}}}$, rebal | $f_{I_{\text{all}}}$, rand DataShare MultiTask |
|---|---|---|---|---|---|
| *lift-any* | 635K | 0.88 | 0.94 | 0.85 | 0.62 |
| *lift-banana* | 9K | 0.04 | 0.13 | 0.38 | 0.09 |
| *lift-bottle* | 11K | 0.02 | 0.16 | 0.66 | 0.15 |
| *lift-sausage* | 5K | 0.02 | 0.10 | 0.38 | 0.15 |
| *lift-milk* | 6K | 0.01 | 0.13 | 0.42 | 0.13 |
| *lift-box* | 6K | 0.00 | 0.12 | 0.16 | 0.08 |
| *lift-can* | 6K | 0.01 | 0.16 | 0.46 | 0.07 |
| *lift-carrot* | 80K | 0.71 | 0.41 | 0.72 | 0.37 |
| *place-any* | 30K | N/A | **0.86** | 0.74 | 0.30 |
| *place-bottom* | 5K | N/A | 0.43 | 0.57 | 0.30 |
| *place-top-right* | 4K | N/A | 0.16 | **0.55** | 0.08 |
| *place-top-left* | 4K | N/A | 0.23 | **0.75** | 0.19 |
| Min | | 0.00 | 0.10 | 0.16 | 0.07 |
| 25-th percentile | | 0.00 | 0.13 | 0.41 | 0.09 |
| Median | | 0.01 | 0.16 | **0.56** | 0.15 |
| Mean | | 0.14 | 0.32 | 0.55 | 0.21 |
| 75-th percentile | | 0.03 | 0.42 | 0.73 | 0.30 |
| Max | | 0.88 | 0.94 | 0.85 | 0.62 |
| Mean (low data) | | 0.01 | 0.18 | 0.42 | 0.13 |

| Task Name | | $f_{I_{\text{all}}}$, rebal | $f_{I_{\text{skill}(1, 0.15)}}$, rebal | $f_{I_{\text{skill}(1, 1)}}$, rand | $f_{I_{\text{skill}(1, 1)}}$, rebal (ours) |
|---|---|---|---|---|---|
| *lift-any* | | **0.95** | 0.80 | 0.88 | 0.89 |
| *lift-banana* | | 0.30 | 0.58 | **0.62** | 0.33 |
| *lift-bottle* | | 0.48 | 0.68 | 0.55 | **0.69** |
| *lift-sausage* | | 0.39 | **0.42** | 0.28 | 0.38 |
| *lift-milk* | | 0.27 | 0.27 | **0.52** | 0.51 |
| *lift-box* | | 0.22 | 0.12 | 0.28 | **0.29** |
| *lift-can* | | 0.28 | **0.47** | 0.43 | 0.43 |
| *lift-carrot* | | **0.75** | 0.52 | 0.71 | 0.70 |
| *place-any* | | 0.24 | 0.83 | 0.57 | 0.85 |
| *place-bottom* | | 0.02 | 0.62 | 0.17 | **0.87** |
| *place-top-right* | | 0.10 | 0.26 | 0.27 | 0.54 |
| *place-top-left* | | 0.16 | 0.39 | 0.22 | 0.53 |
| Min | | 0.02 | 0.12 | 0.17 | **0.29** |
| 25-th percentile | | 0.20 | 0.36 | 0.28 | **0.42** |
| Median | | 0.28 | 0.50 | 0.48 | 0.54 |
| Mean | | 0.35 | 0.49 | 0.46 | **0.58** |
| 75-th percentile | | 0.41 | 0.64 | 0.58 | **0.74** |
| Max | | **0.95** | 0.83 | 0.88 | 0.89 |
| Mean (low data) | | 0.21 | 0.36 | 0.32 | **0.5** |

Table 4: Quantitative evaluation of MT-Opt with different data impersonation and re-balancing strategies. This table reports performance of 7 different models on the 12 ablation tasks, trained on identical offline dataset, with identical computation budget, and evaluated executing 100 attempts for each task for each strategy on the real robots (totaling to 12*100*7=8400 evaluations). In all cases a shared policy for all 12 tasks is learned. The difference across the strategies is in the way the data is impersonated (expanded), and in the way the imperson-ated data is further re-balanced. The last column is our best strategy featuring skill-level data impersonation and further data re-balancing. This strategy outperforms other strategies on many different percentiles across all 12 tasks; however the effect of that strategy is even more pronounced for the tasks having scarce data, e.g. *lift-can*, *lift-box*, *place-top-right*, see Mean (low data) statistic. The column #2 indicates the number of episodes which were collected for each task.

Initially scripted policies were run collecting data for the *lift-anything* and *place-anywhere* tasks. Once performance of our learned policy for these tasks out-performed the scripted policy we shifted to a mix of epsilon greedy and pure on-policy data collection. The majority of our episodes were collected for the *lift-anything* and *place-anywhere* tasks with learned policies. It is worth mentioning that over the course of data collection many good and bad ideas where tried and evaluated via on-policy collection. All of these episodes are included in our dataset. Additional tasks being incorporated over time.

After we had a policy capable of the *lift-anything* and *place-anywhere* tasks we introduced more specific variations of pick and place tasks where either a specific object needed to be picked, or an object needed to be placed in a specific location on the plate. At this point, our data collection process consisted of executing a randomly selected pick task followed by a randomly selected place task.

As a result of the collection process described above, we were left with a 800,000+ episode offline dataset, very diverse along tasks, policies, success rate dimensions.

# 10    Details for real world experiments

The robot workspace setup for the 12 task ablations is shown in Fig 17. Table 4 summarizes studies of 7 different data impersonation and re-balancing strategies for 12 tasks. The last column features the model which on average outperforms other strategies. Note that this strategy is not the best across the board. For example, due to big imbalance of our offline dataset, the native data management strategy (column #3) yields best performance for the over represented tasks, but very bad performance for underrepresented tasks.

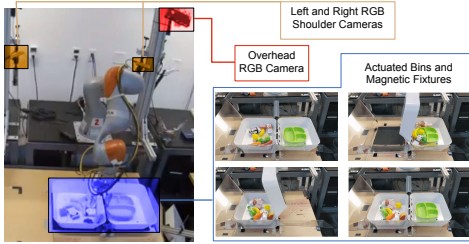

Figure 16: Robot workspace consisting of an overhead camera (red), two over the shoulder cameras (brown), and a pair of articulated resettable bins with a plate fixture that can be magnetically attached to the bin (blue).

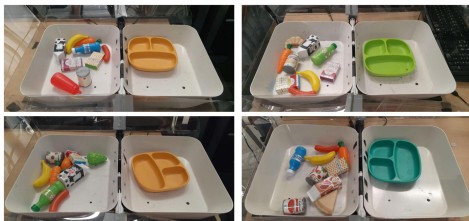

Figure 17: Representative evaluation scenes used for ablation experiments. Contains one of three different color plates. And nine graspable objects: One of each object from our seven object categories with two extra toy food objects sometimes from the seven object categories, sometimes not.

