# OpenReview forum: "Scaling Up Multi-Task Robotic Reinforcement Learning"
_robot-learning.org/CoRL/2021/Conference — CoRL2021 Poster_

### Official Review · Reviewer_k7mA · 2021-07-24

**Originality:** Good
**Technical Quality:** Excellent
**Clarity Of Presentation:** Very Good
**Impact:** 3

**Recommendation:**

Strong Accept: I recommend accepting the paper and will argue for my recommendation even if other reviewers hold a different opinion.

**Summary:**

The paper describes a learning framework Mt-Opt to scale up learning of object arrangement tasks on physical robots in a table top setup.
The paper presents learning and generalization results from data collected over 1.5 years on 7 robots. There are several techniques used to scale up the data collection, from automatic resets, to starting with scripted policies, to redistributing data using known relationships between tasks. The results demonstrate that the framework allows robots to share data and representations, allowing them to learn novel tasks faster.

**Issues:**

1) Results of trying zero shot: What percentage of novel tasks were solved zero-shot if any.
2) Would the data be released?
3) A through list of assumptions about the object arrangement problem being solved. For example resets are available here, which are not a requirement in general object arrangement scenarios.

**Reviewer Expertise:**

Good: General knowledge of the area

**Strengths And Weaknesses:**

Strengths:
1) The success and failure task annotation methods using images is much easier than annotating policies.
2) The length of the robot experiment is impressive, and demonstrates the challenges in learning general purpose policies and representations.
3) The data rebalancing techniques showing that we cannot just dump all available data at the same time.
4) The experiments are very through.

Weakness:
1) This is a lot of data over 7 robots, and 1.5 years. The paper demonstrates sample complexity issues with robot learning. One would expect more zero shot results.
2)  There is significant semantic information present in the data collected. From labelling general and specialized tasks to teaching tasks in the order of first bootstrapping pick tasks from a scripted policy and then learning to place.
3) I am not sure if the learning is faster or more robust when compared to an engineer scripting these pick and place behaviors with conventional vision pipelines such as those available with Cognex.

**Summary Of Recommendation:**

I appreciate the length of the experiment and thoroughness of the results of this paper. The work demonstrates that more fundamental advances in sample complexity and the explore-exploit problems are needed.

---

> ### Author Response · Authors · 2021-08-23
> **Response to Reviewer k7mA**
>
> Thank you for your suggestions, comments and a positive review of our work. We answer the questions below.  We are working to incorporate suggestions into the paper, and will share an updated version as soon as possible.
>
> **Zero shot results**
> We omitted a thorough evaluation of the zero-shot results and only showed examples of candidate tasks that occasionally succeeded in Fig. 5 (which can be further fine-tuned similarly to the cloth covering task). As we mentioned in the response to reviewer xsVH, we will remove these results and use the saved space to discuss the limitations of the system.
>
> **Would the data be released?**
> We are open to releasing all the data collected upon publication.
>
> **List of assumptions**
> Thank you for the suggestion. Please see the list of assumptions of this system below:
> * The tasks that MT-Opt learns are short-horizon tasks ( < 20 time steps). Long horizon real-world robotic tasks are still hard for RL methods.
> * The binary outcome of a task can be determined solely based on the final frame of the episode.
> * All the tasks learned by the MT-Opt policy take advantage of an automated reset mechanism. While this is not an inherent assumption of the method, we have not tried learning any reset-free behaviors.
> * Learning drastically different skills is challenging, as our method works the best at adapting more similar tasks. It remains an open question as to how far MT-Opt can be scaled to a very diverse set of skills.
> * The action space of MT-Opt is 4DoF + toggle gripper action. We found learning higher DoF tasks to be challenging yet principally possible.

---

### Official Review · Reviewer_vENi · 2021-07-26

**Originality:** Good
**Technical Quality:** Excellent
**Clarity Of Presentation:** Excellent
**Impact:** 4

**Recommendation:**

Strong Accept: I recommend accepting the paper and will argue for my recommendation even if other reviewers hold a different opinion.

**Summary:**

This paper proposes a multi-task reinforcement learning framework, which can be considered as a multi-task expansion of QT-Opt. It features an continuous, iterative process of task definition, data collection and RL training. The key components that enable the multi-task data collection and RL training are the success detector, task impersonation and data rebalancing. The success detector along with task impersonation determine what task to collect data for in the next iteration and also whether to include a particular episode from other tasks for training the given task. Data rebalancing makes sure that the number of data samples are balanced across tasks, and the successful/unsuccessful data are balanced within each task.
Through extensive experiments on multiple real world robotic manipulation tasks, the paper showed that the proposed framework significantly improved the success rates across many tasks when compared with baselines that either learn from each individual task or do not use all components in the proposed framework. This framework allows the agent to learn to more efficiently solve harder tasks by bootstrapping from easier tasks. It also makes it possible to define new tasks that are not in the original offline set and leverage data from other tasks to solve the new task. Ablation studies showed that both the data sharing and representation sharing are beneficial.


**Issues:**

Please clarify the items discussed in the weaknesses/suggestions section.

**Reviewer Expertise:**

Very good: Comprehensive knowledge of the area

**Strengths And Weaknesses:**

Strengths
1. This paper is well written and pleasant to read
2. Novel data management strategy that allows scalable data collection and sharing
3. Extensive real-world experiments
4. The accompanying video nicely articulates the key ingredients in the paper

Weaknesses/Suggestions

I find it to be a strong paper with enough details. I don’t have many suggestions to make. Just some clarification questions:
1. Task ID is a one-hot vector. Is it of fixed length? Is it a problem when we are in a continual setting where new tasks are being added?
2. Line 226-227, “allow the individual tasks to be ordered sequentially, where one task is executed after another ...” I’m a bit confused here. Is there some fixed ordering of the tasks? Or is the ordering random?
3. Line 390, “pre-trained MT-Opt policy for collecting the data for the new task”. Is it pre-trained on all 12 tasks, or on the lift-any and place-any tasks as Line 368 seems to suggest?
4. Line 624 in appendix, how is a skill defined and detected?
5. Figure 8, is the input image a concatenation of three grayscale images from the three RGB cameras? If so, it would improve clarity if it’s stated or illustrated.


**Summary Of Recommendation:**

I find it to be a strong system paper that’s relevant to the robot learning community at large and hence I recommend acceptance. The proposed multi-task RL framework is shown to be more effective than the baselines, backed up by large-scale real-world experiments. Although the multi-robots hardware setup might be too expensive for the framework to be widely adopted, it shows the great potential of scaling up existing RL algorithms for multi-task robot manipulations.

---

> ### Author Response · Authors · 2021-08-23
> **Response to Reviewer vENi**
>
> We thank the reviewer for their comments and a positive assessment of our work! Please see our responses below.  We are working to incorporate suggestions into the paper, and will share an updated version as soon as possible.
>
> **Is task ID a fixed length vector?**
> Thank you for the question! We set the task ID vector to be of fixed size that exceeds the number of initial tasks so that it can accommodate more tasks during the process. It allows us to assign an unassigned task ID to a new task as the number of tasks is expanding. We will clarify this aspect of MT-Opt in the manuscript.
>
> **Is the ordering of the tasks fixed?**
> We order placing skills to be executed after lifting skills and allow for all combinations of particular instantionations of these skills such as lifting carrot -> placing in the bottom left corner or lifting sausage -> placing anywhere. This requires a minimal amount of task knowledge and simplifies exploration challenges for the system. We will include this explanation in the paper.
>
> **Is the fine-tuning policy pre-trained on all the tasks or just lift-any and place-any?**
> The policy is pre-trained on all 12 tasks but we use the lift-any and place-any task IDs for the initial exploration of the fine-tuning tasks. We will clarify it in the manuscript.
>
> **How is a skill defined and detected?**
> We assign tasks to different skills by using a very simple mapping: all the lifting tasks are assigned to the lifting skill and all the placing tasks are assigned to the placing skill. Such simple grouping results in significant performance improvement over other data-sharing baselines, which opens up exciting avenues for further data-sharing research in the future.
>
> **Image input architecture**
> The success detection (SD) model has 3 RGB image convolutional towers, while the policy models have 1 RGB image convolutional tower. SD uses 3 camera views, RL policy uses 1 camera view. We found SD learned better from 3 images and this helped policy learning. For the RL policy 3 images caused the policy to train much slower. We will add these specifications to the paper.

---

> > ### Comment · Reviewer_vENi · 2021-09-04
> > **Re: author response**
> >
> > Thank you for your reply to help clarify things

---

### Official Review · Reviewer_xsVH · 2021-07-29

**Originality:** Fair
**Technical Quality:** Good
**Clarity Of Presentation:** Very Good
**Impact:** 3

**Recommendation:**

Weak Reject: I recommend rejecting the paper, but will not argue for my recommendation if the majority of other reviewers have a different opinion.

**Summary:**

This submission desribes MT-Opt, a multi-task robot-learning framework whic
trains a single model to solve all tasks, using semantically driven data
sharing and re-balancing strategies.  The entire framework bootstraps from an
initial collection of labeled human demonstrations, but proceeds by using the
robot attempts as novel demonstrations and training data.

The system is roughly divided into 3 components:  Task definition, which
consists in the labeling of both human and robot demonstrations.  Data
Collection, in which a series of robots is assigned tasks to perform, whose
executions are re-used as new training data.  Data Sharing and RL Training,
which determines how the trajectories which are labeles for a specific task can
be used as training data for similar tasks.

**Issues:**

Major:

* "a policy takes a camera image and a one-hot encoding of the task ..."  How
  can one-hot encodings of non-predefined tasks?  If you extend you task space,
  you have to extend the encoding space.  How does this work?  (I would find no
  issue if a task embedding were to be used, rather than one-hot encoding).
  Similar issue with the predefined probability over tasks in eq.(1).

* What are the details of the data rebalancing?  This seems to be a major part
  of the submission which seems to be not explained.

* The main text mentioned semantic similarity between tasks, but the experiment
  seems lacking.  Aside from the similarity between lift-any and lift-X (for
  any given X), are there other examples?  is there any semantic similarity
  being explored, e.g., between lift-X and lift-Y?  or between lift-X and
  place-X or place-Y?  Which lift- and place- tasks are semantically similar?

* In Fig.7 (and other parts of the text), there is a claim that the system is
  able to perform other novel tasks.  In practice, it seems that these novel
  tasks are simply alternative labels made up by humans, who are allowed to be
  as fancyful as they please since most of these new tasks are never evaluated
  in the real system.

  In fact, one can notice a very distinct difference between the real
  evaluation tasks (only composed of lift-X and place-Y), and the made up
  "novel" tasks (which include very sophisticated tasks such as cover-X,
  serve-y, chase-Z, etc).

  It seems excessive (and mildly misleading) in fig.7 to take this much space
  to show so many pictures of novel tasks when the main take-away is simply
  that robot demonstrations may be reinterpreted as new tasks, despite those
  specific novel tasks not being put to the test.

* Lines 310-312 imply that data re-balancing is only applied to f_I_skill, and
  not f_I_all.  If my understanding is correct, the comparison between
  f_I_skill and f_I_all is extremely unfair, as the re-balancing (rather than
  the data-sharing strategy) may be the important component.

Mid:

* In fig 4, why make such a distinction between pure on-policy (which I assume
  is greedy) and e-greedy policies?  What value epsilon is used?

* In fig.6, are all methods given the same amount of data / experience and/or
  training time?

* In fig.7, top row, how do we know this is actually an intentional
  repositioning action, rather than a failed first grasp which bumps and moves
  the carrot?

Minor:

* In line 31, the sentence "Can we instead amortize the cost of learning this
  repertoire over multiple skills, where the effort needed to learn whole
  repertoire is reduced, easier skills serve to facilitate the acquisition of
  more complex ones, and data requirements, though still high overall, become
  low for each individual behavior?" is a mouthful and hard to parse/read.

* In fig.4, it would help if the order in the legend matches the order in the
  bar plots.

* In the legend of fig.6, what does [36] mean?

* I assume the numbers of episodes in section 5.5 are approximations, since I
  find it hard to believe that the number of successes and failures for any
  given task happens to be clean multiples of 100.  Since all numerical figures
  are provided, I think you should somehow indicate that this is an
  approximation, or simply provide the exact numbers.

* Fig.5 mentions "ablation tasks", which is an unclear description;  as I
  understand, these are the main 12 tasks which the system is primarily trained
  on, so something like "evaluation tasks" would be more suitable.

* "In ... f_I_all, the data is expanded only for the class of tasks having
  similar visuals, dynamics and goals".  How do the tasks differ in visuals and
  dynamics?  Seems like they all have the same visuals and dynamics.  In
  practice, the semantics are only based on the goal, i.e. whether both task
  are lift-X or place-Y.

**Reviewer Expertise:**

Poor: Limited knowledge of the area

**Strengths And Weaknesses:**

Strengths:
  * Addresses a complex problem, very relevant to the robot-learning community.
  * Evaluation is performed on multiple aspects of the system, rather than
    exclusively focusing on the overall performance as a whole.
  * Very well written and pleasant to read.

Weaknesses:
  * While the general outline of the system is clear, a number of key details
    are missing, which makes the submission seem incomplete and to some degree
    non-reproducible.
  * Only suitable for tasks whose success can be assessed by a final
    observation, i.e., cannot handle "execution modalities" or specific ways of
    performing the task (e.g. quickly, or carefully).
  * The work is a bit incremental, as none of the components of MT-Opt seem to
    be particularly original.
  * Some claims seem to be partially embellished or not very close to the
    actual implementation / evaluation details, e.g.:
    * Although the system is described as sharing data belonging to semantically
      similar tasks, in practice the tasks chosen are primarily limited to the
      format lift-X and place-Y, and the semantic sharing seems to be very
      limited, e.g. only between lift-any and lift-X.
    * Although one of the main parts of the motivation was the ability to
      define novel types of tasks, in practice some implementation details made
      use of the pre-known 12 tasks, e.g., using one-hot encodings, even though
      the tasks were not all introduced at the same time.
    * In Fig.1, claims of potential zero-shot learning capabilities are
      unsubstantiated, and the main motivation for the claim is very weak
      (i.e., a human was able to reinterpret a robot demonstration as a novel
      task).

Extensions:
* Tasks being represented as simple prepositions makes me wonder how easy it
  really is to introduce and generalize to new tasks.  A straightforward
  extension would be to consider tasks as predicates which accept arguments,
  e.g., lift(any), lift(carrot), etc.  This way, a new lift-object task would
  not be defined as a completely new task label, but simply as a new object
  representation, paired with an already trained lift() task predicate
  representation.

**Summary Of Recommendation:**

My general impression of the work is positive, however my current
recommendation is a weak reject, in light of certain limitations and issue, not
only with the work itself, but also with its presentation.  Primarily, these
concern missing details and certain embellished claims which make the actual
system seem fancier than it really is (e.g., when it comes to the types of
tasks that it is able to solve, and how tasks can be grouped into semantic
groups, which in practice seems very limited).  To be clear, I think the system
itself is a reasonable (incremental) contribution, in particular given the
great overall difficulty of building such a system to begin with; my main
concern is not the presence of these limitations, but the fact that the text
seems to be trying to sweep these under the rug, in a manner of speaking.  I
think the text should be more honest and open to discussing its own
limitations.

My current recommendation is for the submission as it stands;  I will be happy
to modify it in light of a author response and/or updated manuscript.

---

> ### Author Response · Authors · 2021-08-23
> **Response (1 of 2) to Reviewer xsVH:**
>
> Thank you for your thorough comments and suggestions. Please see the answers below. We are working to incorporate suggestions into the paper, and will share an updated version as soon as possible.
>
> **Fixed embedding space**
>
> Thank you for bringing up this point - we will update the manuscript with the explanation. We start learning with a larger one-hot vector that can accommodate future tasks including tasks that might have not been defined initially. Once a new task is defined and its success detector is trained, we allocate a next available one-hot task ID to this task. For the fine-tuning experiments, we start exploration for the novel tasks by utilizing the most similar task for the initial bootstrap, which is then followed by the on-policy data collection with the task ID of the fine-tuning task.
>
> **Potential zero-shot learning capabilities are unsubstantiated (“a human was able to reinterpret a robot demonstration as a novel task”).**
>
> The learning of the semantic lifting tasks (lift-carrot, lift-bottle, etc.) was bootstrapped from a zero-shot policy trained on relabelled lift-any data.  In part, we additionally substantiate this in Section 5.5 learning from the lift-sausage episodes alone does not work (3% performance trained from 16K episodes).  Running our trained policies slightly out of distribution and reinterpreting the resulting episode is exactly how we bootstrap new tasks! For example, in the “place bottle on a rack” task (Fig 1 bottom left.), we put the rack on the plate so our place-any strategy can occasionally succeed. Once enough episodes are collected, we can retrain the MT-Opt policy and remove the plate entirely. We learn new tasks by reinterpreting and gradually adapting to new tasks. We present an example of such a process in Section 5.6, where we bootstrap towel grasping and object covering policy from a zero-shot generalization of lift-any and place-any policies and with 2 days of data collection yield performance of 92% and 79% on these tasks. Zero-shot learning gives us policies good enough to bootstrap from, which allows us to expand the number of tasks over time.
>
> **The text should be more honest and open to discussing its own limitations**
>
> In Fig. 5, we wanted to express the variety of candidate tasks where MT-Opt policy occasionally succeeded zero-shot (and therefore could be used as other candidates for further fine-tuning, as shown with the cloth-covering example in Sec. 5.6). To address your points, we will remove the Fig. 5 candidate tasks, to make room for more discussion of limitations listed below:
> The tasks that MT-Opt learns are short-horizon tasks ( < 20 time steps). Long horizon real-world robotic tasks are still hard for RL methods.
> The binary outcome of a task can be determined solely based on the final frame of the episode.
> All the tasks learned by the MT-Opt policy take advantage of an automated reset mechanism. While this is not an inherent assumption of the method, we have not tried learning any reset-free behaviors.
> Learning drastically different skills is challenging, as our method works the best at adapting more similar tasks. It remains an open question as to how far MT-Opt can be scaled to a very diverse set of skills.
> The action space of MT-Opt is 4DoF + toggle gripper action. We found learning higher DoF tasks to be challenging yet principally possible.
>
> **What are the details of the data rebalancing?**
>
> We balance the data across the tasks as well as within each task to ensure an equal amount of task successes and failures in a batch. We will add a more detailed description of rebalancing in the text.
>
> **Which lift- and place- tasks are semantically similar?**
>
> Thank you for the question, we will clarify it in the manuscript. We consider all the lifting tasks to be semantically similar resulting in sharing data among the lifting tasks. The same semantic similarity applies to placing tasks that can share data between each other but do not share data with the lifting tasks. The initial and final states for both of these task families differ significantly and we found that sharing data using a different semantic grouping results in a performance worse than the baselines.
>
> **Data re-balancing is only applied to f_I_skill, and not f_I_all.**
>
> Thank you for pointing this out - this is an imprecise statement and we will update the manuscript accordingly. f_I_skill and f_I_all only refer to the data sharing strategy and we ablate the effects of data-rebalancing separately in Table 2. These two aspects are complementary and we achieve the best results by combining re-balancing and f_I_skill data sharing.
>
> **Response continued below:**

---

> > ### Author Response · Authors · 2021-08-23
> > **Response (2 of 2) to Reviewer xsVH**
> >
> > **Continued from above:**
> >
> > **Mid:**
> >
> > **In fig 4, why make such a distinction between pure on-policy (which I assume is greedy) and e-greedy policies? What value epsilon is used?**
> >
> > Pure on-policy refers to running the SOTA policy at the time (epsilon=0). E-greedy uses epsilon = 0.2 (80% on-policy, 20% random exploration for each action). We will include these details in the paper
> >
> > **In fig.6, are all methods given the same amount of data / experience and/or training time?**
> >
> > Same number of training steps, from the same dataset trained at the same point in time. All methods used all data, routed in different ways. The QT-Opt baseline is 12 distinct single task policies each trained using only that task’s data (the sum of which corresponds to all of the data). Given the same data, the difference in performance across different methods is explained solely by the underlying data sharing and rebalancing strategies.
> >
> > **In fig.7, top row, how do we know this is actually an intentional repositioning action, rather than a failed first grasp which bumps and moves the carrot?**
> >
> > The video (https://www.youtube.com/watch?v=H00-gNywtic&t=267s) at 4:20 contains the Fig. 7 sequence. From 420-4:23, the gripper does not close, we consider this a repositioning rather than a lift attempt. The sequence in 4:23-4:24 could be viewed as both.  We would also refer the reviewer to the examples of singulation from QT-Opt (https://ai.googleblog.com/2018/06/scalable-deep-reinforcement-learning.html) which exhibits similar behaviors.
> >
> > **Minor:**
> >
> > **In line 31, the sentence "Can we instead amortize the cost of learning this repertoire over multiple skills, where the effort needed to learn whole repertoire is reduced, easier skills serve to facilitate the acquisition of more complex ones, and data requirements, though still high overall, become low for each individual behavior?" is a mouthful and hard to parse/read.**
> >
> > We will rephrase this sentence to be more clear.
> >
> > **In fig.4, it would help if the order in the legend matches the order in the bar plots.**
> >
> > Thank you! We will address this in the paper.
> >
> > **In the legend of fig.6, what does [36] mean?**
> >
> > This was supposed to be a reference to Qt-OPT. The ordering of our citations changed and we will update the citation number accordingly. Thank you for pointing this out.
> >
> > **I assume the numbers of episodes in section 5.5 are approximations, since I find it hard to believe that the number of successes and failures for any given task happens to be clean multiples of 100. Since all numerical figures are provided, I think you should somehow indicate that this is an approximation, or simply provide the exact numbers.**
> >
> > Yes, these numbers are rounded to the nearest hundred, we will clarify this in the paper.
> >
> > **Fig.5 mentions "ablation tasks", which is an unclear description; as I understand, these are the main 12 tasks which the system is primarily trained on, so something like "evaluation tasks" would be more suitable.**
> >
> > This is a good suggestion, thank you.
> >
> > **"In ... f_I_all, the data is expanded only for the class of tasks having similar visuals, dynamics and goals". How do the tasks differ in visuals and dynamics? Seems like they all have the same visuals and dynamics. In practice, the semantics are only based on the goal, i.e. whether both task are lift-X or place-Y.**
> >
> > The grouping of which tasks belong to a skill is based on the heuristic of which tasks look similar to each other so that multi-tasking can bring the most benefit. In the case of the evaluation tasks, we grouped all the semantic lifting tasks together into one skill and grouped all the placing tasks into a separate skill. This grouping yields the most benefit for this setting but further research is needed to determine how to group tasks into skills automatically.

---

> > > ### Author Response · Authors · 2021-09-03
> > > **Response to the Reviewer**
> > >
> > > Please let us know if our response and the changes in the paper (e.g. adding the limitations section) have addressed your comments. Thank you!

---

### Official Review · Reviewer_UVT2 · 2021-08-01

**Originality:** Good
**Technical Quality:** Very Good
**Clarity Of Presentation:** Good
**Impact:** 3

**Recommendation:**

Weak Accept: I recommend accepting the paper, but will not argue for my recommendation if the majority of other reviewers have a different opinion.

**Summary:**


The authors propose MT-Opt, a method for scaling multi-task robot learning trained on an "arm farm" of real robots. This has two steps: (1) end user specification of the goal state of a task (via demonstrations), and (2) multi-task reinforcement learning.

They train a success detector detector based on these human user demonstrations, then describe a curriculum-like system for collecting additional data, using data from easy tasks to bootstrap harder ones, and rebalancing data to make sure it can learn even these difficult problems. Very importantly, they also use a relabeling strategy so that examples can be used to train multiple tasks simultaneously, which allows them to collect data using these simple policies ("grasp anything on the table", for example), and label it for more specific tasks (grasping a specific object, for example).

**Issues:**


- It seems like the tasks are all very similar, and I'm not sure what the advantage is over existing work (or just using grasp prediction)
- In addition, the method only really applies to RL with clear goals - no explanation of how it could work on dynamic tasks, for example.
- It would be great to explain what the advantages of this method are and how it would scale past the cases show.
- Information about the robot's action space is missing; this makes it hard to tell how well the system actually scales.
- Discussing the applicability of the algorithms to more complex and longer horizon examples would be nice. Could the authors do some simulation experiments on a more complex environment? If the environment is so simple that random pick-and-place can collect positive examples, I think it's hard to make any strong claims about the method.

**Reviewer Expertise:**

Very good: Comprehensive knowledge of the area

**Strengths And Weaknesses:**

On the positive side, the system is able to perform a many tasks in a 2d pick and place environment, and shows both training and task specification that can be scaled on a real robot. Using simple tasks ("pick up anything") to collect data that can then be used to learn a hard task ("pick up the sausage") is a good, simple idea, that I think could be useful for a lot of people.

On the negative side, though, I have a few serious issues with their method. It does not really seem to scale all that well, taking a very large amount of data and being quite restricted in terms of what sorts of tasks it can represent. Basically everything is a 2D pick and place task. And it seems like the tasks are very similar -- "lift banana" vs. "lift sausage," for example. It is not demonstrated on a really wide range of scenarios.

It would be great to compare to transporter nets, another paper that works on a wide range of 2d object pick and place tasks, but which seems to need much less hardware. The high level concept of providing task supervision from human demonstrations has also appeared before, e.g. in Concept2Robot [57]. The authors show doing so in a continuous way, which is interesting.

On the positive side, MT-Opt allows for sharing data between tasks, and they show an improvement by doing so. I thought this was the most interesting part of the paper.

However, this part of the paper is not made very clear. For example, the explanation of "task impersonation" seems very important to what the paper's unique contributions are over the state of the art. However, the explanation is delegated to the appendix. I think it's fine to include lots of details in the appendix, but the paper should still stand on its own. Task impersonation appears to be used to create essentially hard negative examples, which is interesting.

I also like the fact that the success detector can label multiple successes for different tasks at the end of a trial; it seems like they do not need to limit use of the success detector (SD) to episode end, so I wonder if it would change much. Maybe this limitation is due to the training method for the SD? How could the SD generalize to tasks not obvious from a single top-down image? Would this affect the training process?

I also thought this had its weaknesses, though. If the SD is trained online,  couldn't it end up being biased towards a subset of the original data?

Finally, a more philosophical concern: if this approach only works on strongly goal-directed tasks -- those where success or failure is determined by a single image -- why should we use this method, over a method like [48] or transporter nets, which takes advantage of this fact to learn much more efficiently?

Finally, I think the "learn new skill" example was interesting, although the approach seems fairly straightforward: hope that you have a good enough policy that you can collect enough positive examples to start fine-tuning. This again seems not very scalable, since with a higher DOF task space (say, a kitchen), would this even be feasible?


**Summary Of Recommendation:**

I think while the goal of this paper is admirable, we've seen a lot of very high quality results in 2d pick and place tasks, and this paper has enough gaps that I am leaning against publication. Some of the details are still unclear to me, especially what advantages this system has compared to related work. A lot of their contributions (such as using simple tasks to bootstrap harder ones) seem well explored in the literature, and all the tasks are very simple.

In the end, the paper is attempting to solve an important problem, but it shows results using a tremendous amount of time, data, and engineering resources on what seems to be a fairly straightforward suite of tasks, which somewhat undermines their goals.

---

> ### Author Response · Authors · 2021-08-23
> **Response (1 of 2) to Reviewer UVT2**
>
> Thank you for your comments and suggestions. We respond to the questions below.  We are working to incorporate suggestions into the paper, and will share an updated version as soon as possible.
>
> **It is not demonstrated on a really wide range of scenarios**
>
> We would like to clarify that we perform evaluations on random arrangements of the test objects, some of which are drawn randomly for every evaluation run (please see example evaluation scenarios https://mt-opt.github.io/workspace_setups.jpg). Small size of the bin and random configurations of the objects often result in the target object being partially occluded, stuck or difficult to separate from others, requiring the semantic picking tasks to perform additional manipulations to singulate the target object. The test scene is randomly shuffled every episode to ensure that the evaluation is not biased towards a certain object configuration. The scene is set to include the semantic picking objects to enable comparison of multiple methods on the same set of tasks. The system is thoroughly evaluated - over 500 real-robot-hours were spent on evaluations of different variations of the method. The scale of the empirical evaluation for MT-Opt  is substantially larger and more diverse than most works of this sort, e.g. [3, 5, 6, 7, 8]. We believe that the experimental conclusions will be relevant to researchers studying multi-task robotic learning.
>
> **Task impersonation explanation is delegated to the Appendix**
>
> We agree that the task impersonation method is one of the contributions of the paper and we will expand its explanation in the main paper.
>
> **Tasks are similar**
>
> The criteria of what constitutes a separate task is not well defined in robotics but in this work, what separates different tasks is different success conditions. We will clarify this in the paper. Please see the response to the meta-reviewer for the further discussion of the task similarity.
>
> **Advantage over existing work/grasp prediction**
>
> Our goal in this work is to study multi-task robotic learning and less so to engineer the best possible system to perform the tasks presented in the paper. We provide comparisons to other learning approaches, including single-task RL methods (including a very competitive grasping system QT-Opt), other multi-task baselines, as well as a number of ablations, which serve the purpose of studying the multi-task aspects of robotic learning.
>
> **Dynamic tasks**
>
> While MT-Opt in its current form is not able to handle dynamic tasks, it is common for manipulation tasks to be defined in terms of the desired outcome at the last time step. For example, [1, 2] use the terminal time step location of a ball for a ball-in-cup task, [3] use the final position of an object for stacking and relocating, [4] use the final locations of objects for pushing, etc. While there are definitely tasks that cannot be defined in this way, and the current MT-Opt system does not accommodate, for instance, intermediate rewards, we believe that nonetheless this assumption is not very restrictive. Of course, the system could be extended to accommodate learned intermediate rewards as well, using any existing reward learning method. We will however make this explicit in the paper.
>
> **How would the method scale?**
>
> The main advantage of the method is the fact that it can continuously improve and broaden the set of tasks that the robots learn. For our data collection process (described in Section 4), we start with just two tasks, which are, over the course of data collection, scaled up to the 12 evaluation tasks. In Section 5.6, we showcase how a pre-trained MT-Opt model can be used for fast fine-tuning to a task with previously unseen objects to a high performance using just one day of data collection.
>
> **Response Continued Below:**

---

> > ### Author Response · Authors · 2021-08-23
> > **Response (2 of 2) to Reviewer UVT2**
> >
> > **Continued from above:**
> >
> > **Missing action space details**
> >
> > The action space includes a 4 degree-of-freedom (x, y, z and yaw) end-effector position as well as a toggle-gripper and terminate-episode actions. We will include these details in the paper.
> >
> > **More complex, long-horizon tasks**
> >
> > We agree that learning complex, long-horizon tasks is extremely challenging for modern RL methods. We admittedly do not try to tackle this challenge in this paper, instead focusing on data-efficiency gains by multi-task learning.
> >
> > While we have performed multiple simulation experiments that include some results on tasks such as stacking, insertion of box opening, we believe that the community will gain more insights from the real-world results presented in the paper, showcasing the efficiency gains of multi-task RL at scale, which have not been shown before. Adding these additional tasks in the real world would require substantial effort (including many more tasks that bridge the gap between the current tasks and those), which is out of the scope of this paper.
> >
> > **Transporter Nets comparison**
> >
> > Thank you for bringing up this point. Transporter Nets (TN) demonstrate performance on a set of 2D tasks and it is a method with very different assumptions than MT-Opt. In particular, it is an imitation learning method that assumes access to human demonstrations that our system does not require (multi-task SD uses human videos to learn successful frames of a task but these are not used as demonstrations and do not involve teleoperation of robots). The inherent structure of TN limits the applicability of the method to 2D tasks only by recovering a spatially consistent 2D representation from point clouds, whereas in our case, although we present the results on the 4DoF action space, we utilize a general RL method that can be readily extended to more diverse action spaces such as 6DoF. Lastly, TN optimize for a single step trajectory, whereas MT-Opt is able to cope with multi-step RL tasks, which can continuously and autonomously improve, exceeding the performance seen in the initial data collected for such tasks.
> >
> > [1] Kober and Peters, Policy Search for Motor Primitives in Robotics
> >
> > [2] Schwab et al., Simultaneously Learning Vision and Feature-based Control Policies for Real-world Ball-in-a-Cup
> >
> > [3] Riedmiller et al, Learning by Playing, Solving Sparse Reward Tasks from Scratch
> >
> > [4] Finn and Levine, Deep visual foresight for planning robot motion
> >
> > [5] Wulfmeier et al, Regularized Hierarchical Policies for Compositional Transfer in Robotics
> >
> > [6] Singh et al, COG: Connecting New Skills to Past Experience with Offline Reinforcement Learning
> >
> > [7] Zhu et al, The Ingredients of Real World Robotic Reinforcement Learning
> >
> > [8] Kalashnikov et al, Qt-opt: Scalable deep reinforcement learning for vision-based robotic manipulation

---

> > > ### Comment · Reviewer_UVT2 · 2021-08-24
> > > **Quick response to Transporter Nets discussion**
> > >
> > > > The inherent structure of TN limits the applicability of the method to 2D tasks only by recovering a spatially consistent 2D representation from point clouds, whereas in our case, although we present the results on the 4DoF action space, we utilize a general RL method that can be readily extended to more diverse action spaces such as 6DoF. Lastly, TN optimize for a single step trajectory, whereas MT-Opt is able to cope with multi-step RL tasks, which can continuously and autonomously improve, exceeding the performance seen in the initial data collected for such tasks.
> > >
> > > I agree with all your criticisms of transporter nets; the problem I have is that your action space and tasks only seem to show results on tasks that TN could easily handle. Is there evidence for your claims that this can perform better than TN? Is there evidence it can expand to larger action spaces? This was the core problem I had with this paper: lots of big claims, plus 500 hours and 1.5 years of data, and it doesn't seem to do anything we can't already do more easily.
> > >
> > > Additionally, the SD as implemented does not seem like it could handle anything more complex than a goal-directed TN task. This is a major concern for the method as a whole. Am I wrong about this?
> > >
> > > The method seems in principle like it might be able to do more; if there was some evidence, I'd happily change over to a strong accept. My concern with the small action space you used, though, is that it's quite common in RL tasks to do this and it makes the amount of necessary data exponentially smaller (hence my concerns about "scaling" above).

---

> > > > ### Author Response · Authors · 2021-08-26
> > > > **Comparisons with TN**
> > > >
> > > > **As it relates to TN:**
> > > >
> > > > Thank you for raising these points! Transporter Nets (which requires human demonstrations and a calibrated RGBD camera which is not the case for MT-Opt) is a very capable imitation learning system but is based on a different set of principles than MT-Opt which can make drawing a fair comparison between the two difficult. To address your question: "Is there evidence for your claims that this can perform better than TN?", we recorded a number of episodes showcasing a closed-loop control ability of MT-Opt that requires multi-step actions to accomplish the task - these include situations where the object of interest is not visible (or partially occluded) to begin with and the robot needs to perform multiple singulation actions to get to the object first. Please see the additional examples of such behaviors here: https://mt-opt.github.io/closed_loop_behaviors.html.
> > > >
> > > > This is in contrast to TN that inherently decomposes a task into sequences of open-loop pick & place (not allowing for non-prehensile motions like pushing or chasing). In TN, the arm is completely removed from the scene after each action to get the next observation. Another, simpler example of behavior that MT-Opt exhibits that emphasizes this point is the carrot- or bowl- chasing examples in the video here: https://mt-opt.github.io/closed_loop_behaviors.html (Section Carrot-Chasing).
> > > >
> > > > In addition, to make sure that this claim is correct, we also reached out to the authors of the TNs paper with the same examples and they confirmed that their method would struggle to solve them due to the inability to decompose them into sequences of open-loop pick & place.
> > > >
> > > > **As it relates to SD:**
> > > >
> > > > To answer your 2nd question: "the SD as implemented does not seem like it could handle anything more complex than a goal-directed TN task (...) Am I wrong about this?", please consider that the SD provides an abstraction that is beyond a goal image of a pick-and-place task. In the case of MT-Opt, we could easily specify tasks on a different level of abstraction such as "grasp a red object" or "grasp a vegetable" (please see example images at: https://mt-opt.github.io/images/success_detection_capabilities.jpg) which would not be possible by providing a very specific goal image. In addition, one could easily expand the SD with images of open boxes and use it as a success classifier for a box-opening task.
> > > >
> > > > **“The method seems in principle like it might be able to do more; if there was some evidence, I'd happily change over to a strong accept:”**
> > > >
> > > > We hope that with the examples above we have demonstrated the multi-step, closed-loop nature of MT-Opt. These capabilities allow for non-prehensile and recovery behaviors such as searching and singulating to accomplish tasks.

---

> > > > > ### Author Response · Authors · 2021-08-29
> > > > > **Please let us know**
> > > > >
> > > > > Dear reviewer,
> > > > >
> > > > > Please let us know if the additional evidence and the answers above address your concerns. Thank you!

---

> > > > > > ### Comment · Reviewer_UVT2 · 2021-09-03
> > > > > > **Response TN/SD**
> > > > > >
> > > > > > _TN comparison_
> > > > > >
> > > > > > These experiments are nice to see, but doesn't quite prove the point. Carrot-chasing for example seems to be more similar to a sequence of open-loop motions to the carrot, far from a continuous, reactive task execution. But definitely a nice thing to highlight.
> > > > > >
> > > > > > My main concern remains here: yeah, this might be a little more reactive than TN, but we're still mostly just talking about pick and place tasks. The advantages it has are very weak for the tremendous amount of data and extra effort required.
> > > > > >
> > > > > > _SD capabilities_
> > > > > >
> > > > > > The SD still seems quite limited by the fact that it's only a single frame. I like the idea of, e.g., the box-opening task you mention -- it's multi-step and much more interesting than what you show. I'm also not sure how well this method would scale to something like that, though.
> > > > > >
> > > > > > These examples are still goal images; it's nice that the goal image can include the robot state as well. I could certainly see the approach generalizing beyond goal images in the future but it isn't there as far as I can tell.
> > > > > >
> > > > > > _Overall_
> > > > > >
> > > > > > Still conflicted about this paper.

---

> > > > > > > ### Author Response · Authors · 2021-09-03
> > > > > > > **Response to the Reviewer**
> > > > > > >
> > > > > > > Thank you for your response. We appreciate the dialogue and effort that you are putting into evaluating our paper!
> > > > > > >
> > > > > > > **Transporter Nets:**
> > > > > > >
> > > > > > > We encourage you to take a look at other videos as well such as the carrot singulation or work-carrot-out-of-corner examples. In those you can observe how for instance, MT-Opt policy picks up the carrot, but then drops it, re-grasps it closer to the center. This behavior is not possible with TN.
> > > > > > >
> > > > > > > We also like to refer you to the Limitations and Future works section of https://arxiv.org/pdf/2012.03385.pdf, also published by TN authors. They state: “Future work may investigate higher rates of control that learn recovery policies to react in realtime”. This future work is exactly what we demonstrate in MT-Opt.
> > > > > > >
> > > > > > > For further evidence, please take look at Carrot Singulation: video #9.  TN works by sequencing pick and place operations where the arm is removed from the scene in between the episodes - this formulation works well for certain tasks, but does not admit small adjustments to the arm pose in relation to an object. In video #9, the MT-Opt policy will search around in an empty bin if no carrot is visible, while TN would pick up the most carrot-like looking object, a fundamentally different behavior.
> > > > > > >
> > > > > > > **The advantages it has are very weak for the tremendous amount of data and extra effort required.**
> > > > > > >
> > > > > > > We would certainly like to expand the number and kind of tasks. However, we believe that, at this point, this is the most comprehensive evaluation and set of results in *multi-task reinforcement learning in the real-world* (comparable to [1] that has a much simpler setup that includes simple blocks) to-date, and therefore it is a valuable datapoint for the community.
> > > > > > >
> > > > > > > **SD:**
> > > > > > >
> > > > > > > Success detection conditioned on a terminal observation is a common approach in RL [2,3,4,5,6]. Many tasks can be defined this way, but we agree that there are limitations to this approach. For example, care must be taken to ensure that the SD doesn’t break the Markov assumption (i.e. SD from the full episode would require all past observations be provided to the policy as well in order for the task to be fully observable). We are excited about these future directions but we believe this is out of scope for this paper.
> > > > > > >
> > > > > > > It also becomes more challenging to collect SD training data in the real world, if the SD is trained to make a prediction from an entire episode. Since our SD is only trained on terminal images, we were able to very quickly generate lots of positive and negative SD training examples by streaming videos where each video frame is a training example. See MT-Opt video (https://www.youtube.com/watch?v=H00-gNywtic&t=81s) at 1:21
> > > > > > >
> > > > > > > **Box Opening Task:**
> > > > > > >
> > > > > > > While we wanted to focus the paper entirely on the real-world results (since these are the most valued by the CoRL community), we performed multiple tests and ablations in simulation as well including the [box opening task](https://mt-opt.github.io/images/mt-opt-box-open-sim.gif). In order to solve this task in the real world, it would require advancements in reset-free RL - an exciting open area of research. While the community has seen instances of other more complex tasks performed by more specialized methods, we believe that there is value in researching general-purpose systems like MT-Opt and evaluating them in the unstructured settings, even on pick & place tasks.
> > > > > > >
> > > > > > > [1] Riedmiller et al, Learning by Playing, Solving Sparse Reward Tasks from Scratch
> > > > > > >
> > > > > > > [2] Ebert et al, Visual Foresight: Model-Based Deep Reinforcement Learning for Vision-Based Robotic Control
> > > > > > >
> > > > > > > [3] Pinto, Lerrel and Gupta, Abhinav. Supersizing self-supervision: Learning to grasp from 50k tries and 700 robot hours.
> > > > > > >
> > > > > > > [4] Vecerik et al, Leveraging Demonstrations for Deep Reinforcement Learning on Robotics Problems with Sparse Rewards
> > > > > > >
> > > > > > > [5] Xie et al, Few-Shot Goal Inference for Visuomotor Learning and Planning
> > > > > > >
> > > > > > > [6] Fu et al, Variational Inverse Control with Events: A General Framework for Data-Driven Reward Definition

---

> > > > > > > > ### Comment · Reviewer_UVT2 · 2021-09-03
> > > > > > > > **Response**
> > > > > > > >
> > > > > > > > > We encourage you to take a look at other videos as well such as the carrot singulation or work-carrot-out-of-corner examples. In those you can observe how for instance, MT-Opt policy picks up the carrot, but then drops it, re-grasps it closer to the center. This behavior is not possible with TN.
> > > > > > > >
> > > > > > > > TN might not be the best point of comparison here, then, because its own prior work did demonstrate this behavior:
> > > > > > > >   - [push to grasp by Zheng et al](https://ieeexplore.ieee.org/stamp/stamp.jsp?arnumber=8593986&casa_token=ScabLUJuY2YAAAAA:8Ywu6OvWF4c0oZ7zgUwmNqbHA_fkohyp13_bwY0t-d4gG0jHRyQIZIfirEvPlqwUzBd2gbz0Xm0Z3w&tag=1)
> > > > > > > > - [Good robot by Hundt et al](https://ieeexplore.ieee.org/iel7/7083369/7339444/09165109.pdf?casa_token=ngSgw_-BClwAAAAA:F4GLRBhZIKOp1lufaxdVUTGV3ZVxs2j1KxdBlxkv984KCilaPgzv9uxvbXAzAivBjL9jCZ3355km0g)
> > > > > > > >
> > > > > > > > are two that come to mind. The second one I believe does have examples of using the same top-down, simple policy structure, to work objects out of corners given only hours of training -- largely because of the simpler action space this style of approach can use.
> > > > > > > >
> > > > > > > > > We would certainly like to expand the number and kind of tasks. However, we believe that, at this point, this is the most comprehensive evaluation and set of results in multi-task reinforcement learning in the real-world
> > > > > > > >
> > > > > > > > I can appreciate this. I think one thing I dislike about the way the paper was initially written is, it feels like it overstates how useful this technique would be on its own.
> > > > > > > >
> > > > > > > > > It also becomes more challenging to collect SD training data in the real world, if the SD is trained to make a prediction from an entire episode.
> > > > > > > >
> > > > > > > > This is exactly my concern about the current SD approach.
> > > > > > > >
> > > > > > > > > Box Opening Task
> > > > > > > >
> > > > > > > > I do actually quite like this result, although I can see why it was not included in the paper.
> > > > > > > >
> > > > > > > > I think after the discussion and some thought I'll bump my review up to weak accept.

---

> > > > > > > > > ### Author Response · Authors · 2021-09-03
> > > > > > > > > **Thank you!**
> > > > > > > > >
> > > > > > > > > Thank you for your comments and the discussion! We very much appreciate the score change as well. We'll take your comments into account to further improve the paper.

---

> > ### Comment · Reviewer_UVT2 · 2021-09-03
> > **Final comments**
> >
> > I think I have most of the same concerns as before. As one of the other reviewers note, I think the authors are really overselling what they can accomplish here, especially given how resource- and data-intensive this method appears to be.
> >
> > > Scalability
> >
> > The comments on scalability just don't make much sense to me still. You're showing tasks that are a subset of a task you've already learned (grasp anything), and that in a very constrained environment. This doesn't show ability to scale to, say, a precise manipulation task.

---

### Author Response · Authors · 2021-08-23
**Response to everyone**

We thank the reviewers for their time. We are working to incorporate suggestions into the paper, and will share an updated version as soon as possible.

The meta-review states, “There is no takeaway/insight in this work(...) the paper fails to provide its technical contribution”. To clarify the insights and contributions, we state them in bullet points form below and we are reworking the introduction to make them more explicit.  We strongly feel that these contributions merit publication, and that the community would benefit from the insights surfaced at scale and validated on real hardware in this work.

**MT-Opt technical contributions:**
-  **MT-Opt system: A novel scalable framework for learning new skills and tasks:** This is a systems paper that describes a scalable multi-task reinforcement learning system. It  integrates approaches that have been presented in prior work, such as success detection using classifiers or continuous Q-learning, as well as new approaches such as novel data sharing and re-balancing techniques and methods for learning new tasks quickly via multi-task pre-training and fine-tuning on real hardware. We show how different aspects of multi-task learning have a positive impact on the data-efficiency compared to single-task approaches - a result that might seem obvious in retrospect, but has not been demonstrated on a real multi-task RL system at scale in any prior work.
- **Task impersonation and data balancing:** Sharing and re-balancing data yields significant benefits when done with care (harmful if done naively), which was not demonstrated in the multi-task RL literature before. We carefully explore and ablate the effects of different design choices in that space.

**Insights/takeaways from MT-Opt:**
- While some of the individual components MT-Opt uses have been demonstrated before, this is the first work that shows how the combination of the following multi-task components contributes to a unified multi-task RL framework: success detector leverages data from other tasks (success classifiers have been shown in previous works such as [1] but have not been integrated into an offline, model-free RL system to enable data-sharing), sharing weights helps with learning tasks more efficiently (which has only been shown in the real world, but only in simplified settings such as [2]), multi-task policies help with exploration for existing and new tasks (also demonstrated in [2] but in an idealized setting), careful data-sharing between RL tasks significantly improves the results across all tasks (which hasn’t been shown in the real-world multi-task robotic learning setting).
- Large, multi-task robotic datasets can be leveraged via offline RL to learn new tasks in a data efficient manner as long as the new tasks are “close” to existing tasks. In multi-task learning the number of “close” tasks grows as new tasks are learned. To the best of our knowledge, this is the first work that demonstrates this expansion of tasks over time (as described in the 2nd part of Section 4 and 5.5) as well as fast fine-tuning to new tasks using multi-task RL (described and evaluated in Section 5.6).
- Data-sharing in multi-task RL can be effective as long as the tasks sharing data are similar. Putting all of the multi-task data in a single replay buffer is far from optimal, the system needs to carefully balance and share the data to be effective. This is the first work to demonstrate this phenomenon on real robotic tasks.
- The results show that scaling multi-task RL to many tasks is challenging as the tasks need to be similar to gain advantage from data-sharing and bootstrapping and the expansion of tasks takes data and time. This outcome encourages future research on data-sharing techniques for tasks that are further apart.

We demonstrate these insights and contributions through thorough real-world ablations (using over 500 real-robot hours of experiments) of various data sharing strategies and other aspects of the system.

[1] Ebert et al, Visual Foresight: Model-Based Deep Reinforcement Learning for Vision-Based Robotic Control

[2] Riedmiller et al, Learning by Playing, Solving Sparse Reward Tasks from Scratch

---

> ### Author Response · Authors · 2021-08-27
> **Updated paper to incorporate reviewer feedback.**
>
> Dear Reviewers,
>
> We’ve updated the paper to reflect the requested changes (highlighted in red) including improved clarity, listing limitations of the system (Sec. 5.7), details regarding task impersonation contribution (Algorithm 1 and Sec. 3.2), among others. Please let us know if these changes are satisfactory and if you have any other questions or requests. Thank you for helping us improve our paper!

---

### Meta-Review · Area_Chair_B9qB · 2021-08-13

**Recommendation:** Accept (Poster)
**Confidence:** 5

**Metareview:**

Summary: This paper presents a system for multi-task robot learning at scale that combines several existing ideas such as success classifiers etc. Although the individual components and insights are not novel, the overall system has clearly involved significant thought and work.

However, AC and Reviewers find several issues with the paper:
1. Multi-task: Is it really?  In the experimental setup, the tasks seem quite arbitrary. What makes a task different from a different task? It is unclear why lift-banana needs to be a different task than lift-carrot. Shouldn't 'lift' be the task and different objects are just different instances of lifting. Overall, the definition has been chosen just to show lots of different tasks. AC finds this nomenclature misleading and agrees with reviewers there is lot of embellishment and projection of fanciness.
2. Furthermore since both picking and lifting require grasping, it would be nice to comparisons against state of art grasping methods. This well help address if multi-task robot learning is really necessary to learn how to perform such simplistic grasping behaviors compared to off the shelf controller+planners.
3. There is no takeaway/singular insight in this work. As acknowledged in the introduction and related works section, most of the components have been studied in detail before. Success classifiers have been used in several prior work, most notably in work by Ebert and Finn [1], where success classifiers were used in combination with model-based RL. Unfortunately, the paper fails to provide its technical contribution at all.
4. Applicability to dynamic tasks
5. Reproducibility: Given the scale of data used for the project and the arbitrary amount of data collected + mode of collection, I wonder if this experiment can be ever reproduced. Will the authors release all the data?


===

Post Rebuttal:

There was substantial discussion between reviewers and authors.  AC still finds issue with the task definition. Lifting a carrot and banana might be different but they are just different instance of same category (Note two chairs also look different but they are still part of same category called chair). AC believes the paper claims much more than it actually achieves.

AC is convinced that there is definitely value in building systems even though individual ideas might have existed. AC is highly appreciative of the hardware experiments while lot of people only focus on simulation. Overall, AC is inclined to accept but would highly recommend authors to include discussion on tasks, have more baselines and even boil down some of the claims [some of them seem over the board frankly].

In terms of reproducibility: the author promise release of data...the AC would suggest releasing all code as well so people with same robot can reproduce the results [including low-level code if possible].

---

> ### Author Response · Authors · 2021-08-23
> **Response to meta-reviewer**
>
> We thank the meta-reviewer for summarizing the reviews and pointing out various questions and suggestions on how to improve the paper.   We are working to incorporate suggestions into the paper, and will share an updated version as soon as possible.
>
> We would first emphasize that, while the work has limitations (which we discuss below), we also believe that it is going to be of great interest to the research community. The overwhelming majority of robotic reinforcement learning research is conducted in simulation, and it remains an open question how reinforcement learning methods can scale to real-world settings and what challenges they encounter there. In supervised learning domains (e.g., vision), large-scale training on large datasets has proven very effective. Can the same hold in robotics, and what challenges do we have to overcome to get there? Our work doesn't answer these questions fully, but it represents a step in this direction. Making progress on real-world reinforcement learning is very hard, because it requires an extensive investment of time and effort, but such systems have a lot of promise for future robotic control. This kind of progress necessarily requires system building, as does all large-scale ML, and papers focusing on system building will necessarily need to sacrifice in other areas.
>
> We respond to the more specific points raised by the meta-reviewer below:
>
> 1. **Multi-task:**
> It is an important and valid question to ask what constitutes distinct tasks or not. Different tasks in our system correspond to different success conditions. While mechanically lifting a carrot is similar to lifting a banana, a policy that aims to find and lift a carrot will not succeed at lifting a banana. Both the perception and the action components matter in this regard, and our system perhaps places more emphasis on the perception side of the equation. That said, we emphasize that the downstream evaluation tasks do exhibit considerable variability, as shown in Figure 5. We agree, of course, that there are many classes of interesting manipulation behaviors that are not included, and we will revise the discussion section to make it clear that one limitation of our evaluation is that our tasks differ much more in appearance rather than physical motion. We will also revise the paper to clarify what constitutes different tasks in this setting.
> Based on our analysis (see Section 5.4), we find that tasks that are similar benefit from the multi-task setting (very distinct tasks do not). While the task set grows over time allowing for more and more tasks to be similar, it requires a considerable amount of time, which caused us to focus on a set of different but similar tasks, where multitasking is effective. We will add this discussion to the paper.
>
> 2. **Comparison to classical methods:**
> We compare our method to QT-Opt, which is a notably strong learning-based grasping approach. We view the comparison to non-RL-based approaches to be out of scope.
> The main goals of the paper surround a study on how simultaneously learning multiple tasks can improve performance and efficiency compared to single-task reinforcement learning. We do not claim that this is the SOTA approach for the evaluation tasks presented, hence a comparison to non-learning-based approaches is out of the scope of the stated claims and contributions of the paper. We believe that the empirical study in this paper that focuses on comparison of different learning-based approaches still provides a valuable contribution to the robot learning community.
>
> 3. **Insight/takeaway of this work:**
> We would like to emphasize that while this paper presents a few novel algorithmic improvements (e.g. task impersonation and its influence on multi-task training), its main contribution is a multi-task robotic reinforcement learning system that is evaluated at scale in the real world. We put together a list of insights that the robot learning community can draw from this system’s paper in the general reply to all the reviewers.
>
> 4. **Dynamic tasks:**
> While MT-Opt in its current form is not able to handle dynamic tasks, a sparse reward at the last timestep is a common assumption taken in many robot learning approaches [cite]. We address this question in more detail in the response to Reviewer UVT2 under the section Dynamic Tasks.
>
> 5. **Reproducibility:**
> To enable reproducibility as much as possible, we will release all the data upon publication.

---

### Decision · Program_Chairs · 2021-09-13

**Decision:**

Accept (Poster)

**Comment:**

Summary: This paper presents a system for multi-task robot learning at scale that combines several existing ideas such as success classifiers etc. Although the individual components and insights are not novel, the overall system has clearly involved significant thought and work.

However, AC and Reviewers find several issues with the paper:
1. Multi-task: Is it really?  In the experimental setup, the tasks seem quite arbitrary. What makes a task different from a different task? It is unclear why lift-banana needs to be a different task than lift-carrot. Shouldn't 'lift' be the task and different objects are just different instances of lifting. Overall, the definition has been chosen just to show lots of different tasks. AC finds this nomenclature misleading and agrees with reviewers there is lot of embellishment and projection of fanciness.
2. Furthermore since both picking and lifting require grasping, it would be nice to comparisons against state of art grasping methods. This well help address if multi-task robot learning is really necessary to learn how to perform such simplistic grasping behaviors compared to off the shelf controller+planners.
3. There is no takeaway/singular insight in this work. As acknowledged in the introduction and related works section, most of the components have been studied in detail before. Success classifiers have been used in several prior work, most notably in work by Ebert and Finn [1], where success classifiers were used in combination with model-based RL. Unfortunately, the paper fails to provide its technical contribution at all.
4. Applicability to dynamic tasks
5. Reproducibility: Given the scale of data used for the project and the arbitrary amount of data collected + mode of collection, I wonder if this experiment can be ever reproduced. Will the authors release all the data?


===

Post Rebuttal:

There was substantial discussion between reviewers and authors.  AC still finds issue with the task definition. Lifting a carrot and banana might be different but they are just different instance of same category (Note two chairs also look different but they are still part of same category called chair). AC believes the paper claims much more than it actually achieves.

AC is convinced that there is definitely value in building systems even though individual ideas might have existed. AC is highly appreciative of the hardware experiments while lot of people only focus on simulation. Overall, AC is inclined to accept but would highly recommend authors to include discussion on tasks, have more baselines and even boil down some of the claims [some of them seem over the board frankly].

In terms of reproducibility: the author promise release of data...the AC would suggest releasing all code as well so people with same robot can reproduce the results [including low-level code if possible].